# Mean-Shift PCA by Knockoff Mean

**Mengda Li**[1]  **Zeng Li**[2]  **Jianfeng Yao**[1]

## Abstract

Removing noise is difficult, but adding noise is easy. In this work, we show how to eliminate mean-shift noisy components from PCA by deliberately introducing knockoff mean-shift perturbation. Standard PCA is highly sensitive to shifts in the sample mean: a small fraction of samples from a shifted distribution can cause large deviations in the leading principal components. In high-dimensional regimes, existing Robust PCA approaches cannot handle the mean-shift contamination structure inherent in the mixture model. Using tools from Random Matrix Theory, we prove that the mean-shift spikes are spectrally separable from the stable eigenvalues of the original covariance. Furthermore, the original eigenspace remains asymptotically invariant to the contamination, independent of the mixture weight. Exploiting this spectral stability, we propose a simple, two-stage PCA algorithm by adding knockoff mean that identifies and removes the mean-shift component using only standard PCA operations.

## 1. Introduction

High-dimensional data analysis faces inherent challenges due to the curse of dimensionality, necessitating robust dimensionality reduction techniques. Principal Component Analysis (PCA) has served as a cornerstone of multivariate statistics for decades, enabling efficient data representation and dimension reduction. Despite its widespread adoption, PCA exhibits a critical, well-documented sensitivity to violations of its core assumption—that data is centered around a single mean. This fragility is severely exposed in the prevalent setting of *mean-shift contamination*, where observations are drawn from a mixture of a primary distribution

[1]School of Data Science, The Chinese University of Hong Kong, Shenzhen, China [2]Department of Statistics and Data Science, Southern University of Science and Technology, Shenzhen, China. Correspondence to: Zeng Li <liz9@sustech.edu.cn>, Jianfeng Yao <jeffyao@cuhk.edu.cn>.

*Proceedings of the 43rd International Conference on Machine Learning*, Seoul, South Korea. PMLR 306, 2026. Copyright 2026 by the author(s).

and a secondary, homoscedastic component with a shifted mean. In such a regime, *even a small fraction of contaminating samples can systematically bias the empirical mean and, consequently, distort the recovered principal components away from the true covariance eigenvectors.*

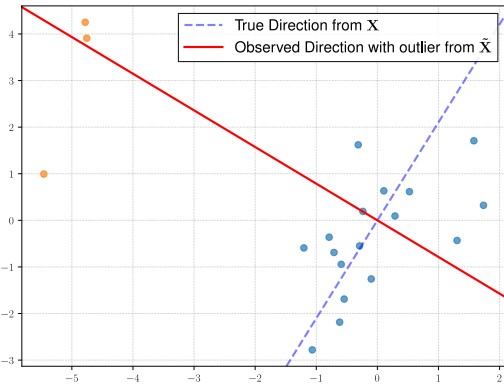

*Figure 1.* **Failure of Classical PCA** on Gaussian data with one mean-shift cluster. Data points (blue) are sampled from a 2D Gaussian mixture with two components: an inlier component centered at the origin (blue) and an outlier component with a mean shift (orange). The red line indicates the first principal component estimated by standard PCA, which is biased towards the outlier cluster and almost orthogonal to the first principal component (blue dashed line) of the uncontaminated data. $d/n = 0.1$, outlier proportion $\pi_1 = 15\%$.

The vulnerability of standard PCA to systematic mean shifts presents a significant obstacle in modern machine learning pipelines, where data is frequently aggregated from multiple sources or contains uncurated batches. Despite numerous Robust PCA (RPCA) variants proposed in the literature (Candès et al., 2011; Wright et al., 2009; Zhou et al., 2010; Xu et al., 2012; Netrapalli et al., 2014; Cai et al., 2019; 2021; Jambulapati et al., 2020; Paul et al., 2024), standard PCA remains the dominant choice in practical applications due to its widespread implementation in statistical software packages. However, *existing RPCA methods fail to recover the true principal components in the mean-shift contamination mixture models in high-dimensional regimes.* The cosine similarity between the recovered low-rank components and

the true eigenvectors converges to zero as the aspect ratio of dimension to sample size increases.

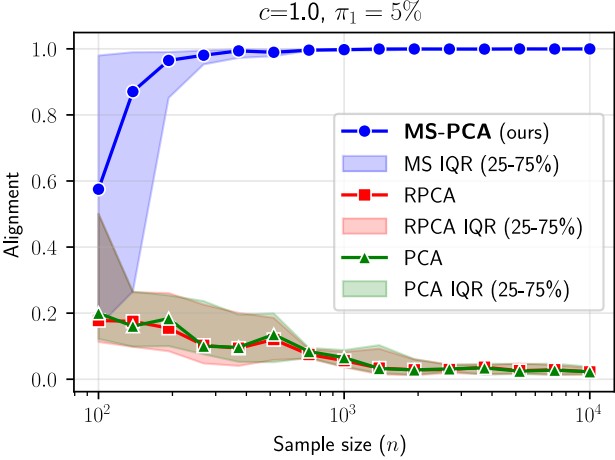

*Figure 2.* **Failure of Robust PCA in high dimensions with only** $5\%$ **noisy samples** Largest principal component cosine alignment of PCA methods on Gaussian data with one mean-shift cluster of outlier proportion $\pi_1 = 5\%$. The Robust PCA method fails to recover the true principal component as the dimension increases w.r.t. non-vanishing aspect ratio $d/n = 1$, while our Mean-Shift PCA consistently recovers the true component. Other settings are described in Figure 6.

In the scenario of first-moment contamination, RPCA methods are fundamentally mismatched. The mean-shift noise is *not sparse*, contradicting the core sparsity assumption of RPCA. Moreover, it exhibits a *low-rank structure* which falls under the same hypothesis as the true signal. Consequently, RPCA formulations do not handle mean-shift contamination well, as illustrated in Figure 2, and this misalignment is exacerbated in high dimensions.

In this work, we use the tools of Random Matrix Theory (RMT) to show that most principal components remain stable in a high-dimensional mean-shift mixture. Furthermore, we show that it is possible to only use *vanilla PCA* to identify and eliminate the contamination-induced components when contamination occurs in the first moment. RMT is essential in this regime because the aspect ratio $d/n$ does not vanish: sample eigenvalues and eigenvectors exhibit deterministic high-dimensional bias and phase-transition behavior. This allows us to distinguish the constant-order spectral displacement caused by an added knockoff mean from the $\mathcal{O}(n^{-1/2})$ fluctuations of stable covariance-induced spikes.

**Problem Formulation**   Our objective is to recover the principal components of the original, uncontaminated data matrix $\mathbf{X}_n \in \mathbb{R}^{d \times n}$ from the observed $(k+1)$-component mean-shift mixture $\widetilde{\mathbf{X}}_n \in \mathbb{R}^{d \times n}$.

The original data consist of $n$ i.i.d. samples arranged column-

wise:

$$\mathbf{X}_n = \big[\mathbf{x}_{(1)}, \ldots, \mathbf{x}_{(n)}\big]_{d \times n}, \qquad \mathbf{x}_{(i)} \in \mathbb{R}^d.$$

The contaminated matrix $\widetilde{\mathbf{X}}_n$ is obtained by adding a structured mean-shift matrix $\mathbf{A}_n$:

$$\widetilde{\mathbf{X}}_n = \mathbf{X}_n + \mathbf{A}_n, \qquad \mathbf{A}_n = \sum_{i=1}^{k} \mathbf{m}_{(i)} \boldsymbol{\gamma}_{(i)}^{\top}. \quad (1)$$

The $i$-th subpopulation is characterized by a mean vector $\mathbf{m}_{(i)} \in \mathbb{R}^d$ and a mixture weight $\pi_i \in (0, 1)$. Its membership is encoded by a binary indicator vector $\boldsymbol{\gamma}_{(i)} \in \{0,1\}^n$, where $(\boldsymbol{\gamma}_{(i)})_j = 1$ if and only if the $j$-th sample originates from the $i$-th subpopulation.

The model therefore comprises a centered inlier component (with $\mathbf{m}_{(0)} = \mathbf{0}$) and $k$ outlier components with distinct mean shifts.

*Remark* 1.1.   This formulation is commonly referred to as a signal-plus-noise model in the literature, where $\mathbf{A}_n$ typically represents a low-rank signal matrix and $\mathbf{X}_n$ corresponds to noise. However, in our context, the roles are reversed: the low-rank matrix $\mathbf{A}_n$ constitutes the mean-shift contamination that we aim to eliminate, or at least eliminate its influence with respect to the covariance structure of the underlying data $\mathbf{X}_n$.

**Contribution**   We provide a precise characterization of PCA's behavior under mean-shift mixture using RMT. Based on the additive low-rank random perturbation model (Benaych-Georges & Nadakuditi, 2012), we show that contamination from a shifted subpopulation induces an *asymptotically orthogonal distortion* to the sample covariance matrix, leaving the original covariance eigenspace invariant. Our main contributions are:

- **Perturbation-Based Correction:** We introduce a 2-fold PCA algorithm that recovers the stable covariance structure by strategically introducing another artificial mean-shift perturbation to detect non-stable principal components induced by a mean-shift mixture. This procedure is effective when the mixture weight is sufficiently prominent.

- **Invariant Principal Components:** We show that there exists an asymptotic invariance of the underlying covariance eigenspace of general data with compactly supported spectrum, i.e., most of the principal components of the original data are stable, regardless of the mean-shift mixture weight.

**Paper Organization**   The paper is organized as follows: Section 1.1 surveys relevant literature and positions our contribution within existing work. The proposed Mean-Shift

PCA algorithm is stated in Section 2. Our main theoretical results, including mathematical assumptions for each model and the RMT-based recovery guarantees, are presented in Section 3. Section 4 provides numerical validation compared to robust PCA methods, robust covariance estimators, and simple preprocessing baselines. Finally, Section 5 discusses broader implications, limitations, and potential extensions of our work.

## 1.1. Related Work

**Principal Component Pursuit**  Robust Principal Component Analysis aims to decompose the data matrix into a low-rank signal component $\mathbf{L}$ and a sparse outlier component $\mathbf{S}$. The original RPCA formulation proposed by Candès et al. (2011) solves:

$$
\begin{aligned}
\min_{\mathbf{L},\mathbf{S}} \quad & \|\mathbf{L}\|_* + \alpha\|\mathbf{S}\|_1 \\
\text{subject to} \quad & \widetilde{\mathbf{X}} = \mathbf{L} + \mathbf{S},
\end{aligned}
\tag{2}
$$

where $\|\cdot\|_*$ denotes the nuclear norm, $\|\cdot\|_1$ is the vectorwise $\ell_1$-norm, and $\alpha > 0$ is a regularization parameter typically set to $\alpha = 1/\sqrt{\max(d, n)}$.

**Stable PCP** (Zhou et al., 2010) relaxes the equality constraint in (2) as $\|\widetilde{\mathbf{X}} - \mathbf{L} - \mathbf{S}\|_F \leq \delta$ to account for small dense noise $\mathbf{N} = \widetilde{\mathbf{X}} - \mathbf{L} - \mathbf{S}$, where $\delta > 0$ bounds the Frobenius norm of the noise component. **Outlier Pursuit** (Xu et al., 2012) modifies the $\ell_1$-norm in (2) to the $\ell_{1,2}$-norm to promote column-wise sparsity in $\mathbf{S}$.

More recent work (Cai et al., 2019) formulates the RPCA problem with explicit rank and sparsity constraints:

$$
\begin{aligned}
\min_{\mathbf{L},\mathbf{S}} \quad & \|\widetilde{\mathbf{X}} - \mathbf{L} - \mathbf{S}\|_F \\
\text{subject to} \quad & \text{rank}(\mathbf{L}) \leq r, \ \|\mathbf{S}\|_0 \leq s,
\end{aligned}
\tag{3}
$$

for some pre-specified rank $r$ and sparsity level $s$.

However, none of these RPCA formulations can effectively handle the mean-shift contamination structure inherent in the mean-shift mixture model. The resulting low-rank components in $\mathbf{L}$ poorly approximate the true covariance eigenspace (largest principal components of $\mathbf{X}$), as quantified by the vanishing cosine similarity in high dimensions.

**$\ell_1$-PCA**  A prominent approach to robust PCA replaces the $\ell_2$-norm with the $\ell_1$-norm in the optimization objective to enhance resistance to outliers. Kwak (2008) proposed an $\ell_1$-norm maximization formulation for PCA:

$$
\mathbf{W}^* = \arg\max_{\mathbf{W}} |\mathbf{W}^\top \widetilde{\mathbf{X}}|_1 \quad \text{subject to} \quad \mathbf{W}^\top \mathbf{W} = \mathbf{I}_m,
$$

which aims to find the projection direction $\mathbf{W}^*$ maximizing the $\ell_1$-norm of the projected data. This formulation

reduces the influence of outliers due to the linear growth of the $\ell_1$-norm compared to the quadratic growth of the $\ell_2$-norm. Subsequent advances (Markopoulos et al., 2014; 2017) developed more efficient algorithms for $\ell_1$-PCA.

This formulation, however, leads to non-convex optimization problems that are, in general, NP-hard; their solutions usually converge to local minima. Existing algorithms exhibit exponential complexity in the feature dimension $d$, rendering them *computationally prohibitive for high-dimensional data*.

**Median-of-Means PCA**  Paul et al. (2024) propose Median-of-Means PCA (MoMPCA), which views PCA as empirical risk minimization over rank-$r$ projection matrices and replaces the usual average reconstruction loss by the median of blockwise losses. The method first centers the data using a robust location estimator, partitions the samples into blocks, and optimizes the median-of-means reconstruction objective by projected gradient updates on an orthonormal basis. In finite-dimensional implementations, however, MoMPCA remains an iterative non-convex optimization method whose per-iteration cost is $O(d^2b + dr^2 + L)$ where $b$ is the block size, and $L$ is the number of blocks.

**Robust Sub-Gaussian PCA**  Jambulapati et al. (2020) study PCA under an $\epsilon$-corruption model for centered sub-Gaussian distributions and provide the first polynomial-time algorithms that recover nontrivial covariance information under sub-Gaussian assumptions alone. Their filtering algorithm repeatedly computes a top direction of a weighted empirical covariance, estimates the variance in that direction robustly, and downweights samples with large projected mass; it returns a $(1 - O(\epsilon \log \epsilon^{-1}))$-approximate top eigenvector with $O(\frac{nd^2}{\varepsilon} \log \frac{n}{\varepsilon} \log n)$ time complexity. They also develop a width-independent Schatten-$p$ packing algorithm, achieving nearly-linear dependence on the input size under a spectral-gap condition, but at the cost of a weaker approximation factor and polynomial dependence on $\epsilon^{-1}$.

## 2. Methodology

**Remove Mean-Shift Effect via Mean-Shift Data Augmentation**  Adding noise to the data is always easier than removing it. Extra distortion of the noise can be a friend in finding the truth (Daras et al., 2023). Rather than attempting to eliminate mean-shift outliers directly, our algorithm strategically introduces additional mean-shift contamination to modify the spectral signature of the existing noise. This manipulation causes the spiked eigenvalues associated with contamination to shift, while leaving those corresponding to the true covariance structure invariant.

The key observation is that while contamination-induced eigenvalues will migrate under additional noise injection,

the eigenvalues originating from the uncontaminated data remain stable. This differential response provides a mechanism to distinguish between noise artifacts and genuine spectral components. The algorithm then selectively retains only those eigenvalues and eigenvectors demonstrating invariance—corresponding to the true underlying covariance structure. We present our procedure in Algorithm 1.

**Notation 1** (Spectral Separation). *In high dimensions, spiked eigenvalues outside the support of the limiting spectral measure separate into two disjoint sets:*

- $\Lambda_{\mathbf{A}}$ *is determined by the first moment (mean)*
- $\Lambda_{\mathbf{P}}$ *is determined by covariance and higher-order moments*

*Adding an artificial knockoff mean-shift to the data perturbs only $\Lambda_{\mathbf{A}}$, leaving $\Lambda_{\mathbf{P}}$ unchanged. These two sets are defined precisely in Theorem 3.5.*

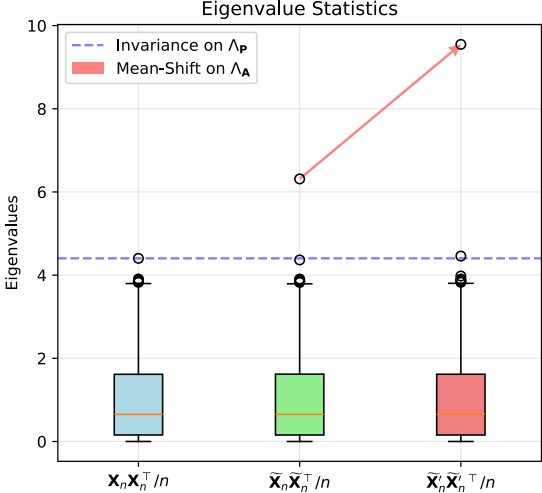

*Figure 3.* **Knockoff Mean Impact**: Boxplot on the eigenvalues of $\mathbf{X}_n \mathbf{X}_n^\top / n$, $\widetilde{\mathbf{X}}_n \widetilde{\mathbf{X}}_n^\top / n$, and $\widetilde{\mathbf{X}}'_n (\widetilde{\mathbf{X}}'_n)^\top / n$. The additional perturbation $\mathbf{A}'_n$ induces shifts in the spiked eigenvalue(s) in $\Lambda_{\mathbf{A}}$ associated with mean-shift contamination, while leaving the stable eigenvalue(s) of the original covariance structure in $\Lambda_{\mathbf{P}}$ invariant. Data drawn from a 2-component Gaussian mixture. $d/n = 1, n = 10^3, \pi_1 = 50\%, \ell_1 = 2\sqrt{c}, \theta_1^2 = 4\sqrt{c}$.

**Perturbation Generation** The additional perturbation $\mathbf{A}'_n$ can be generated using random vectors $\mathbf{m}'$ and $\boldsymbol{\gamma}'$. The direction of knockoff mean $\mathbf{m}'$ may be sampled uniformly from the unit sphere $\mathbb{S}^{d-1}$ or from an i.i.d. Gaussian distribution. The mixture weight $\pi' \in (0, 1]$ associated with $\boldsymbol{\gamma}'$ can be chosen arbitrarily large (e.g., $\pi' = 0.5$ or even $\pi' = 1$) such that the artificial perturbation strength $\theta'^2 := \pi' \|\mathbf{m}'\|^2 > 1/D_{\mu_\infty}(\lambda^+ + \epsilon)$ (Remark C.2) is sufficiently strong to induce detectable spectral shifts among the

largest eigenvalues.

In practice, we recommend choosing $\theta'^2$ to be comparable in magnitude to the underlying representation $\theta^2$ of the observed spiked eigenvalues we aim to identify. For example, to target the largest observed eigenvalue $\tilde{\lambda}_1$, one may set

$$\theta'^2 = 2g^{-1}(\tilde{\lambda}_1),$$

where $g^{-1}$ is the inverse of the spike-forward mapping $g$, characterized by Proposition C.1. Ideally, the function $g$ should be replaced by $D_{\mu_\infty}$ for general compactly supported LSD, provided that $D_{\mu_\infty}^{-1}$ is computable.

Several edge cases are worth noting. If the original mean-shift strength lies below the BBP threshold, it does not create an outlying spectral spike, so MS-PCA is neutral and standard PCA already sees no detectable mean-shift spike to remove. If a covariance spike and a mean-shift spike are nearly coincident, the knockoff perturbation still displaces only the mean-shift component while the covariance-induced spike remains stable. The same reasoning applies when the artificial direction happens to align with an existing mean-shift direction: the singular values of $\mathbf{A}_n + \mathbf{A}'_n$ change by constant order, whereas the covariance-induced spikes remain within their $\mathcal{O}(n^{-1/2})$ fluctuation scale.

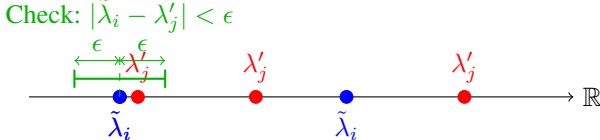

*Figure 4.* **Invariance check**: For each original eigenvalue $\tilde{\lambda}_i$ (blue), we check if there exists a perturbed eigenvalue $\lambda'_j$ (red) within distance $\epsilon = Cn^{-1/2}$. The $\epsilon$-interval is shown for one $\tilde{\lambda}_i$.

**Order of Fluctuation** Without altering the spectra of $\mathbf{A}_n$, the spiked eigenvalues induced by the mean-shift spikes in $\Lambda_{\mathbf{A}}$ exhibit normal fluctuations of order $\mathcal{O}(n^{-1/2})$ (Benaych-Georges & Nadakuditi, 2012, Theorem 2.19). Similarly, eigenvalues in $\Lambda_{\mathbf{P}}$ also fluctuate at the $\mathcal{O}(n^{-1/2})$ scale (Couillet & Hachem, 2013; Couillet & Liao, 2022, Theorem 2.16). The smaller the eigenvalue is, the smaller order of fluctuation it has. The largest eigenvalues at the edge of the limiting spectral distribution (LSD) experience fluctuations of order $\mathcal{O}(n^{-2/3})$ (Baik et al., 2005; Couillet & Liao, 2022, Theorem 2.15), which is the largest fluctuation order for eigenvalues within the support of LSD.

Consequently, to distinguish the mean-shift-induced spikes, we set the threshold $\epsilon$ in Algorithm 1 to scale as $\epsilon = Cn^{-1/2}$ for some constant $C > 0$. Our eigenvalue matching procedure focuses exclusively on the spiked eigenvalues lying outside the support of LSD. We do not distinguish between

**Algorithm 1** Mean-Shift PCA (MS-PCA)

1: **Input:** Contaminated data $\widetilde{\mathbf{X}}_n$, threshold constant $C$
2: **Initial PCA:** Compute the (largest spiked) eigenvalues $\{\tilde{\lambda}_i\}$ and eigenvectors $\{\tilde{\mathbf{u}}_i\}$ of $\widetilde{\mathbf{X}}_n\widetilde{\mathbf{X}}_n^\top/n$.
3: **Noise Injection:** Generate knockoff mean $\mathbf{m}'$ and indicator $\boldsymbol{\gamma}'$ with weight $\pi'$. Form the mean-shift contamination matrix $\mathbf{A}'_n = \mathbf{m}'\boldsymbol{\gamma}'^\top$ and the doubly perturbed data matrix $\widetilde{\mathbf{X}}'_n = \widetilde{\mathbf{X}}_n + \mathbf{A}'_n$.
4: **Second PCA:** Compute eigenvalues $\{\lambda'_i\}$ and eigenvectors $\{\mathbf{u}'_i\}$ of $\widetilde{\mathbf{X}}'_n(\widetilde{\mathbf{X}}'_n)^\top/n$.
5: **Invariance Check:** For each initially observed spiked eigenvalue $\tilde{\lambda}_i$, check if there exists a corresponding $\lambda'_j$ such that

$$|\tilde{\lambda}_i - \lambda'_j| < \epsilon, \quad \text{where } \epsilon = Cn^{-1/2}. \qquad (4)$$

Remove non-stable eigenspaces of $\widetilde{\mathbf{X}}_n\widetilde{\mathbf{X}}_n^\top/n$ and output the stable eigenspaces associated with $\tilde{\lambda}_i$ satisfying Equation 4.

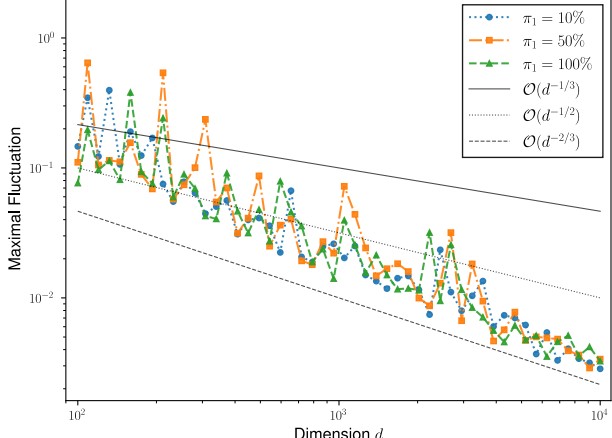

*Figure 5.* **Fluctuation Order**: Maximal fluctuation of stable eigenvalues i.e., $\max_i |\tilde{\lambda}_i - \lambda'_i|$ for $\tilde{\lambda}_i, \lambda'_i$ not in the neighborhood of $\Lambda_{\mathbf{A}}, \Lambda'_{\mathbf{A}}$, versus dimension $d$ for varying contamination mixture weight $\pi_1$ on Gaussian data. The observed decay aligns with the $\mathcal{O}(n^{-1/2})$ threshold in Algorithm 1 (condition 4). $c = d/n = 1$.

eigenvalues in the bulk, which are closely spaced and generally not among the top principal components. Empirically (see Figure 5), $C = 1$ suffices when $d$ is sufficiently large. For moderate $d$ and small aspect ratio $c < 1$, we choose $C = 1/c$ to compensate the weaker concentration of measure in low dimensions.

**Time Complexity** MS-PCA requires two partial PCA computations and one eigenvalue matching step. With a Lanczos or randomized SVD implementation, computing the top $K$ components of a $d \times n$ data matrix costs $O(Knd)$. Since the number of detectable spikes is finite in our asymptotic model, $K = O(1)$ and the leading cost of MS-PCA is $O(nd)$ which is much lower than that of modern robust estimators (Jambulapati et al., 2020; Paul et al., 2024). The matching step is negligible for finite $K$. Appendix E reports empirical runtimes, showing that the extra PCA pass remains substantially cheaper than the optimization-based robust PCA and robust covariance alternatives tested here.

## 3. Theoretical Results

In this section, we present our main theoretical results on PCA's robustness to mean-shift contamination in high-dimensional settings. Our analysis starts with spiked covariance models and is later extended to distributions with compactly supported spectra. First, we establish the phase transition threshold for the largest eigenvalues in the sample covariance matrix of the contaminated data, and for those induced by the mean-shift noise. Next, we characterize the largest eigenvectors and show the asymptotic orthogonality between the eigenspaces corresponding to the intrinsic

covariance structure and those induced by the mean-shift contamination.

To establish a unified framework, we first posit a core assumption regarding the mean-shift contamination structure, shared by all models under study. We assume the mean-shift contamination is aligned with independent random vectors $\mathbf{v}_{(i)}$, and the contamination occurs through independent uniform sampling across outlier subpopulations:

**Assumption 3.1** (Mean-Shift Mixture). The mean-shift matrix $\mathbf{A}_n \in \mathbb{R}^{d \times n}$ admits a finite-rank decomposition: $\mathbf{A}_n = \sum_{i=1}^k \mathbf{m}_{(i)}\boldsymbol{\gamma}_{(i)}^\top$ with $\text{rank}(\mathbf{A}_n) = k$. The mean-shift directional vectors $\mathbf{v}_{(i)} := \mathbf{m}_{(i)}/\|\mathbf{m}_{(i)}\|$ are independently distributed. Each $\mathbf{v}_{(i)}$ is either:

- uniformly distributed on the unit sphere $\mathbb{S}^{d-1}$ as Haar-distributed column vector, or

- with i.i.d. entries of zero mean, $1/d$ variance and satisfying a log-Sobolev inequality (Benaych-Georges et al., 2011).

The binary membership vectors $\boldsymbol{\gamma}_{(i)} \in \{0,1\}^n$ satisfy $\sum_{i=0}^k \boldsymbol{\gamma}_{(i)} = \mathbf{1}_n$, and the mixture weights are defined as $\pi_i = \frac{1}{n}\sum_{j=1}^n \boldsymbol{\gamma}_{(i)j}$ for $i = 0, \ldots, k$, s.t. $\sum_{i=0}^k \pi_i = 1$.

The magnitudes $\|\mathbf{m}_{(i)}\|$ and mixture weights $\pi_i$ are considered deterministic. The mean-shift directions $\{\mathbf{v}_{(i)}\}_{i=1}^k$ are independent of the membership indicators $\{\boldsymbol{\gamma}_{(i)}\}_{i=1}^k$, and the resulting mean-shift matrix $\mathbf{A}_n$ is independent of the original uncontaminated data matrix $\mathbf{X}_n$.

---

**Algorithm 2** Parameter Selection for MS-PCA

---

1: **Signal strength $\theta'$:** Choose $\theta'$ by the inverse spike-forward mapping, e.g., $\theta'^2 = 2g^{-1}(\tilde{\lambda}_1)$, to create a detectable spike comparable to the observed one.
2: **Mixture weight $\pi'$:** Choose a large artificial mixture weight such as $\pi' = 0.5$ or $\pi' = 1$.
3: **Direction of $\mathbf{m}'$:** Sample a random unit vector uniformly from the sphere $\mathbb{S}^{d-1}$, or normalize a vector with i.i.d. Gaussian entries.
4: **Magnitude of $\mathbf{m}'$:** Set $\|\mathbf{m}'\|^2 = \theta'^2/\pi'$.
5: **Threshold:** Set $\epsilon = Cn^{-1/2}$. In our experiments, $C = 1$ works well when $d$ is large, while $C = 1/c$ is used for smaller aspect ratios.

---

**Notation 2.** *We can decompose $\mathbf{A}_n$ as:* $\mathbf{A}_n = \sqrt{n}\mathbf{V}_n\boldsymbol{\Theta}\mathbf{W}_n^\top$ *where* $\mathbf{V}_n = \left( (\mathbf{v}_{(i)})_{i=1}^k \right) \in \mathbb{R}^{d\times k}$, $\boldsymbol{\Theta} = \mathrm{diag}\,(\theta_i)_{i=1}^k$ *with* $\theta_i = \sqrt{\pi_i}\|\mathbf{m}_{(i)}\|$, *and* $\mathbf{W}_n = \left( (\boldsymbol{\gamma}_{(i)}/\sqrt{n\pi_i})_{i=1}^k \right) \in \mathbb{R}^{n\times k}$.

We introduce the Stieltjes transform for asymptotic eigenvalue characterization:

**Definition 3.2.** The Stieltjes transform of a measure $\mu$ is defined as:

$$S_\mu(z) = \int \frac{1}{t-z}d\mu(t)$$

### 3.1. Mean-Shifted Spiked Covariance Model

In this section, we primarily focus on the Gaussian mixture model. The uncontaminated data matrix $\mathbf{X}_n \in \mathbb{R}^{d\times n}$ can be expressed as.

$$\mathbf{X}_n = \boldsymbol{\Sigma}^{\frac{1}{2}}\mathbf{Z}_n, \tag{5}$$

where $\mathbf{Z}_n$ consists of i.i.d. $\mathcal{N}(\mathbf{0}, \mathbf{I}_d)$ columns.

Although the Gaussian case suffices to reveal the fundamental phenomena, we consider a more general setting where $\mathbf{Z}_n \in \mathbb{R}^{d\times n}$ is a random matrix with independent and identically distributed (i.i.d.) entries satisfying the following assumption:

**Assumption 3.3.** $\mathbf{Z}_n \in \mathbb{R}^{d\times n}$ is a random matrix with i.i.d. entries of zero mean, unit variance and finite fourth moment. The ratio between the ambient dimension $d$ and the sample size $n$ converges to a constant $c \in (0, \infty)$ asymptotically: $\frac{d}{n} \to c$.

In practical applications, data often exhibits a low intrinsic dimensionality $r \ll d$, embedded within the high-dimensional ambient space $\mathbb{R}^d$, i.e. the finite-rank signal component lives with isotropic noise in the ambient space. We will hence discuss the spiked covariance model case:

**Assumption 3.4.** The population covariance matrix admits the decomposition:

$$\boldsymbol{\Sigma} = \mathbf{I}_d + \mathbf{P} \tag{6}$$

The matrix $\mathbf{P} \in \mathbb{R}^{d\times d}$ is symmetric with finite rank $r$, and its non-zero eigenvalues are denoted by $(\ell_i)_{i=1}^r$. Furthermore, $(\mathbf{I}_d + \mathbf{P})$ is non-singular.

For simplicity of exposition, we assume later all eigenvalues $\ell_i$ are positive, which aligns with most practical applications. The case of negative eigenvalues can be handled analogously in (Baik & Silverstein, 2006; Couillet & Liao, 2022, Remark 2.13).

#### 3.1.1. EIGENVALUE PHASE TRANSITION AND SPIKE INDEPENDENCE

In this section, we state the most interesting phenomenon: spike independence for the spiked eigenvalues of the sample covariance matrix of the contaminated data $\widetilde{\mathbf{X}}_n$ in (1). In short, among the $r + k$ largest eigenvalues of the sample covariance matrix $\widetilde{\mathbf{X}}_n\widetilde{\mathbf{X}}_n^\top/n$, $r$ eigenvalues are determined by the population covariance spectra $(\ell_i)_{i=1}^r$ while the other $k$ eigenvalues are determined by the mean-shift spectra $(\theta_i)_{i=1}^k$. Crucially, these two sets of spikes evolve independently— neither group influences the other asymptotically.

**Theorem 3.5** (Largest Eigenvalues). *Under Assumptions 3.1, 3.3 and 3.4, supposing all $\ell_i$'s are positive, let $\tilde{\lambda}_1 \geq \tilde{\lambda}_2 \geq \cdots \geq \tilde{\lambda}_{r+k}$ be the $r + k$ largest eigenvalues of the sample covariance matrix $\widetilde{\mathbf{X}}_n\widetilde{\mathbf{X}}_n^\top/n$, and let $\Lambda$ be the set of the asymptotic largest spiked eigenvalues defined as:*

$$\Lambda_\mathbf{P} := \left\{ 1 + \ell_i + c\frac{1+\ell_i}{\ell_i} \,\middle|\, \ell_i > \sqrt{c},\ i \in [r] \right\}$$

$$\Lambda_\mathbf{A} := \left\{ 1 + \theta_j^2 + c\frac{1+\theta_j^2}{\theta_j^2} \,\middle|\, \theta_j^2 > \sqrt{c}, j \in [k] \right\}$$

$$\Lambda := \Lambda_\mathbf{P} \bigcup \Lambda_\mathbf{A}$$

*Then, denoting $\lambda_i^*$ as the $i$-th largest element in $\Lambda$,*

$$
\begin{aligned}
\tilde{\lambda}_i &\xrightarrow[a.s.]{n\to\infty} \lambda_i^*, && \text{if } i \leq \mathrm{card}(\Lambda), \\
\tilde{\lambda}_i &\xrightarrow[a.s.]{n\to\infty} (1+\sqrt{c})^2, && \text{else}
\end{aligned}
\tag{7}
$$

Let $[r] := \{1, 2, \ldots, r\}$ denote the set of the first $r$ non-zero natural numbers. The value $(1 + \sqrt{c})^2$ is the upper edge

of the Marčenko-Pastur law (Marčenko & Pastur, 1967) for aspect ratio $c$. For general distribution with compactly supported spectra (Section 3.2), it is replaced by the supremum of the support of LSD.

*Remark* 3.6. As evident from the definition of $\Lambda$, the spiked eigenvalues of the sample covariance matrix decompose into two asymptotically independent sets: one determined by the population covariance spikes $(\ell_i)_{i=1}^r$, and the other by the mean-shift contamination spikes $(\theta_i)_{i=1}^k$. Consequently, perturbing either set of parameters does not affect the other set of eigenvalues in the asymptotic limit. This spectral decoupling implies that the mean-shift contamination and the intrinsic covariance structure are asymptotically independent at the level of the spiked eigenvalues. Our proposed algorithm leverages this phenomenon by intentionally introducing additional structured mean-shift noise to manipulate the spectral signature of $\mathbf{A}_n$, thereby distinguishing noise-induced spikes from those from uncontaminated data to recover the true covariance structure.

**Corollary 3.7.** *Under the framework of Theorem 3.5, let $\Lambda'$, $\Lambda'_{\mathbf{A}}$ and $\Lambda'_{\mathbf{P}}$ be the set of the asymptotic largest spiked eigenvalues of the sample covariance matrix $\widetilde{\mathbf{X}}'_n(\widetilde{\mathbf{X}}'_n)^\top/n$ after additional mean-shift contamination in Algorithm 1. Then, for sufficiently strong perturbation $\mathbf{A}'_n$, we have:*

$$\Lambda'_{\mathbf{P}} = \Lambda_{\mathbf{P}}, \qquad \Lambda'_{\mathbf{A}} \neq \Lambda_{\mathbf{A}}$$

*Proof.* This result follows directly from Theorem 3.5. The uncontaminated data spikes remain invariant under additional mean-shift contamination, while the contamination-induced spikes shift due to the change of the singular value decomposition in the mean-shift matrix (from $\mathbf{A}_n$ to $\mathbf{A}_n + \mathbf{A}'_n$) after artificial perturbation. $\square$

To establish Theorem 3.5, we state a more theoretical lemma characterizing the asymptotic behavior of spiked eigenvalues via Stieltjes transform.

**Notation 3.** *Denote by $S_c(z)$ the Stieltjes transform of the Marčenko-Pastur law with aspect ratio $c$:*

$$S_c(z) = \frac{-[z + (c-1)] + \sqrt{(z-\lambda^+)(z-\lambda^-)}}{2zc}$$

*where $\lambda^{\pm} = (1 \pm \sqrt{c})^2$ are the upper and lower edges of its support.*

**Lemma 3.8.** *Under Assumptions 3.1, 3.3 and 3.4, let $\tilde{\lambda}^{(n)}$ be a non-zero spiked eigenvalue of the sample covariance matrix lying outside the bulk spectrum i.e., $\tilde{\lambda}^{(n)} \notin [(1-\sqrt{c})^2, (1+\sqrt{c})^2] \cup \{0\}$ and $\det\left(\widetilde{\mathbf{X}}_n\widetilde{\mathbf{X}}_n^\top/n - \tilde{\lambda}^{(n)}\mathbf{I}_d\right) = 0$, then there exists $i \in [r]$ or $j \in [k]$ such that one of the*

*following conditions holds almost surely asymptotically:*

$$1 + \frac{\ell_i}{1+\ell_i}\tilde{\lambda}^{(n)}S_c(\tilde{\lambda}^{(n)}) \xrightarrow[a.s.]{n\to\infty} 0, \qquad (8)$$

$$1/\theta_j^2 - \tilde{\lambda}^{(n)}S_c(\tilde{\lambda}^{(n)})S_{1/c}(\tilde{\lambda}^{(n)}) \xrightarrow[a.s.]{n\to\infty} 0, \qquad (9)$$

*Remark* 3.9. The condition in (8) is equivalent to saying that $\lambda$ is a spiked eigenvalue from the sample covariance matrix of the uncontaminated data $\mathbf{X}_n\mathbf{X}_n^\top/n$, induced by the population covariance spike $\ell_i$, as established in (Baik & Silverstein, 2006). The condition in (9) characterizes the spiked eigenvalues induced by the mean-shift contamination spikes $\theta_i$.

### 3.2. General Data in Mean-Shift Mixture

We now extend our analysis to general data distributions with compactly supported spectra, beyond the spiked covariance model. In this section, we present a general theorem on eigenvectors of uncontaminated data that subsumes the spiked covariance model. To facilitate theoretical analysis, we introduce the following notation for the empirical spectral distribution (ESD) and the limiting spectral distribution:

**Notation 4.** *Let $\mu_n$ denote the ESD of the sample covariance matrix of uncontaminated data $\mathbf{X}_n\mathbf{X}_n^\top/n$, defined by*

$$\mu_n := \frac{1}{d}\sum_{i=1}^d \delta_{\lambda_i(\mathbf{X}_n\mathbf{X}_n^\top/n)}$$

*and let $\mu'_n$ denote the ESD of $\mathbf{X}_n^\top\mathbf{X}_n/n$. Furthermore, let $\mu_\infty$ and $\mu'_\infty$ represent the corresponding limiting spectral distributions of $\mu_n$ and $\mu'_n$ respectively.*

Consider the following assumption on the spectral distribution of uncontaminated data:

**Assumption 3.10.** The ESD $\mu_n$ of $\mathbf{X}_n\mathbf{X}_n^\top/n$ converges weakly almost surely to a non-random compactly supported probability measure $\mu_\infty$ as $d/n \xrightarrow[n\to\infty]{} c \in (0,\infty)$.

Under random mean-shift contamination, the covariance structure of data has an important property of spectral invariance in the high-dimensional regime. Specifically, an eigenvector of the uncontaminated sample covariance matrix $\mathbf{X}_n\mathbf{X}_n^\top/n$ remains asymptotically an eigenvector of the contaminated matrix $\widetilde{\mathbf{X}}_n\widetilde{\mathbf{X}}_n^\top/n$, preserving its associated eigenvalue:

**Theorem 3.11** (Eigenspace Invariance). *Under Assumptions 3.1 and 3.10, let $\mathbf{u}$ be an eigenvector of the uncontaminated sample covariance matrix $\mathbf{X}_n\mathbf{X}_n^\top/n$ associated with eigenvalue $\lambda$, i.e., $\frac{1}{n}\mathbf{X}_n\mathbf{X}_n^\top\mathbf{u} = \lambda\mathbf{u}$, then*

$$\frac{1}{n}\widetilde{\mathbf{X}}_n\widetilde{\mathbf{X}}_n^\top\mathbf{u} = \lambda\mathbf{u} + \mathbf{r}_n, \quad with \quad \|\mathbf{r}_n\|_2 = \mathcal{O}_p(n^{-1/2})$$

*This asymptotic equivalence is denoted by: $\frac{1}{n}\widetilde{\mathbf{X}}_n\widetilde{\mathbf{X}}_n^\top\mathbf{u} \sim \lambda\mathbf{u}$. Roughly speaking $\mathbf{u}_\lambda \sim \tilde{\mathbf{u}}_\lambda$ for $\lambda \in Spec(\mathbf{X}_n\mathbf{X}_n^\top/n)$.*

*Table 1.* **Eigenvector Residual Order**: Maximal residual norm $\|\frac{1}{n}\mathbf{X}_n\mathbf{X}_n^\top\tilde{\mathbf{u}} - \tilde{\lambda}\tilde{\mathbf{u}}\|_2$ for eigenvectors associated with stable eigenvalues (for $\tilde{\lambda}$ not in the neighborhood of $\Lambda_\mathbf{A}$) on Gaussian data. Values are presented with two decimal places. $n = d, c = 1$.

| $\pi_1$ | $d = 1000$ | $d = 10000$ | $d = 50000$ |
|---|---|---|---|
| 0.1% | $1.50 \times 10^{-1}$ | $5.20 \times 10^{-2}$ | $2.37 \times 10^{-2}$ |
| 1% | $1.49 \times 10^{-1}$ | $5.48 \times 10^{-2}$ | $2.88 \times 10^{-2}$ |
| 10% | $1.50 \times 10^{-1}$ | $4.84 \times 10^{-2}$ | $2.50 \times 10^{-2}$ |
| 30% | $1.57 \times 10^{-1}$ | $4.88 \times 10^{-2}$ | $2.55 \times 10^{-2}$ |
| 50% | $1.60 \times 10^{-1}$ | $5.30 \times 10^{-2}$ | $2.58 \times 10^{-2}$ |
| 100% | $1.32 \times 10^{-1}$ | $5.26 \times 10^{-2}$ | $2.75 \times 10^{-2}$ |

This invariance property, combined with the characterization of mean-shift-induced spectral spikes in equation 9 provides the theoretical foundation for our algorithm to distinguish genuine principal components from contamination artifacts.

## 4. Experiment

Based on our theoretical results, we conduct experiments to validate the eigenvector invariance property stated in Theorem 3.11 for stable principal components satisfying condition 4.

**Two-Component Gaussian Mixture with One-Spike Population Covariance**  We generate data from a two-component Gaussian mixture model whose population covariance matrix contains one spike. The population covariance matrix contains one single spike at $\ell_1 = 2\sqrt{c}$. The mean-shift contamination vector is configured with norm $\|\mathbf{m}_{(1)}\| = 2\sqrt{\sqrt{c}/\pi_1}$ where $\pi_1$ is the mixture weight of noise. The direction of $\mathbf{m}_{(1)}$ is sampled uniformly from the unit sphere $\mathbb{S}^{d-1}$ via the QR decomposition of a random i.i.d. Gaussian matrix.

In real-world settings, we do not have access to the uncontaminated eigenvectors $\mathbf{u}$ used in Theorem 3.11. Therefore, our statistical estimator for the uncontaminated eigenspace consists of the contaminated eigenvectors associated with stable eigenpairs $(\tilde{\lambda}, \tilde{\mathbf{u}})$ that satisfy the stability condition 4 in Algorithm 1. We evaluate the residual alignment of the uncontaminated sample covariance matrix $\|\frac{1}{n}\mathbf{X}_n\mathbf{X}_n^\top\tilde{\mathbf{u}} - \tilde{\lambda}\tilde{\mathbf{u}}\|_2$ for the contaminated eigenvectors $\tilde{\mathbf{u}}$ associated with stable eigenvalues. The results in Table 1 show that the residual alignment decays as $\mathcal{O}(n^{-1/2})$, regardless of the contamination proportion $\pi_1$. This decay rate matches that of the other residual $\|\frac{1}{n}\widetilde{\mathbf{X}}_n\widetilde{\mathbf{X}}_n^\top\mathbf{u} - \lambda\mathbf{u}\|_2$ for uncontaminated eigenvectors $\mathbf{u}$ of the contaminated sample covariance matrix, as predicted by Theorem 3.11. This suggests that even contaminated eigenspaces approximately retain the invariance property in high-dimensional settings.

**Comparison to Robust PCA approach**  We compare our Mean-Shift PCA (MS-PCA) algorithm to the modern implementation of Robust PCA (RPCA) via Accelerated Alternating Projections (AAP) introduced by Cai et al. (2019) for recovering the largest principal component in the Gaussian setting. We vary the contamination proportion $\pi_1$ from $5\%$ to $50\%$ and the aspect ratio $c = d/n$ from 0.1 to 1. We measure the eigenvector alignment between the estimated principal component $\hat{\mathbf{u}}_1$ and the true principal component $\mathbf{u}_1$ defined as $|\langle\mathbf{u}_1, \hat{\mathbf{u}}_1\rangle|$. For the parameter settings, we set $C = 1/c, \pi' = 1, \theta'^2 := \pi'\|\mathbf{m}'\|^2 = 2g^{-1}(\tilde{\lambda}_1)$ for MS-PCA and the rank parameter $r = 1$ for RPCA. As illustrated in Figure 6, MS-PCA significantly outperforms RPCA in high-dimensional settings. When $d$ is large, RPCA fails to recover the true principal component and the eigenvector alignment converges to $0$. Moreover, in the relatively low-dimensional setting ($c = 0.1$), our method achieves perfect recovery of the true principal component when the contamination proportion is sufficiently large (e.g., $\pi_1 \geq 25\%$). A larger $\pi_1$ means more samples are shifted by the knockoff mean, which in turn makes the second perturbation more pronounced and improves the algorithm's ability to identify stable eigenvalues.

**Comparison to Additional Robust Estimators**  We further compare MS-PCA with robust covariance estimators and preprocessing baselines with stable implementations: Tyler's M estimator (Tyler, 1987), Huber's M estimator, $\ell_1$-PCA, winsorized-PCA, and center-PCA. The robust PCA methods of (Jambulapati et al., 2020; Paul et al., 2024) are omitted due to the lack of public Python implementations. The setting matches the Gaussian experiment above with $d = 900$, $n = 10^3$, and the same parameter choices as Figure 6. Table 2 reports alignment over 200 trials. MS-PCA stays above $95\%$, while alternatives remain close to random alignment.

These results also explain why classical robust covariance methods underperform here. They are designed mainly for fixed- or low-dimensional regimes, while for $d/n \to c > 0$, sample eigenvectors have non-negligible high-dimensional bias (Johnstone & Paul, 2018). MS-PCA instead uses the RMT spike mapping and the separation between $\mathcal{O}(1)$ mean-shift displacement and $\mathcal{O}(n^{-1/2})$ stable-spike fluctuation. Appendix E reports an additional mean-shift plus covariance-shift experiment, where MS-PCA again gives the strongest recovery.

## 5. Conclusion

A robust estimator for principal components under contamination in high dimensions is a challenging problem. Non-robustness arises for the $K$-largest principal components when $K$ is small, as PCA naturally becomes more

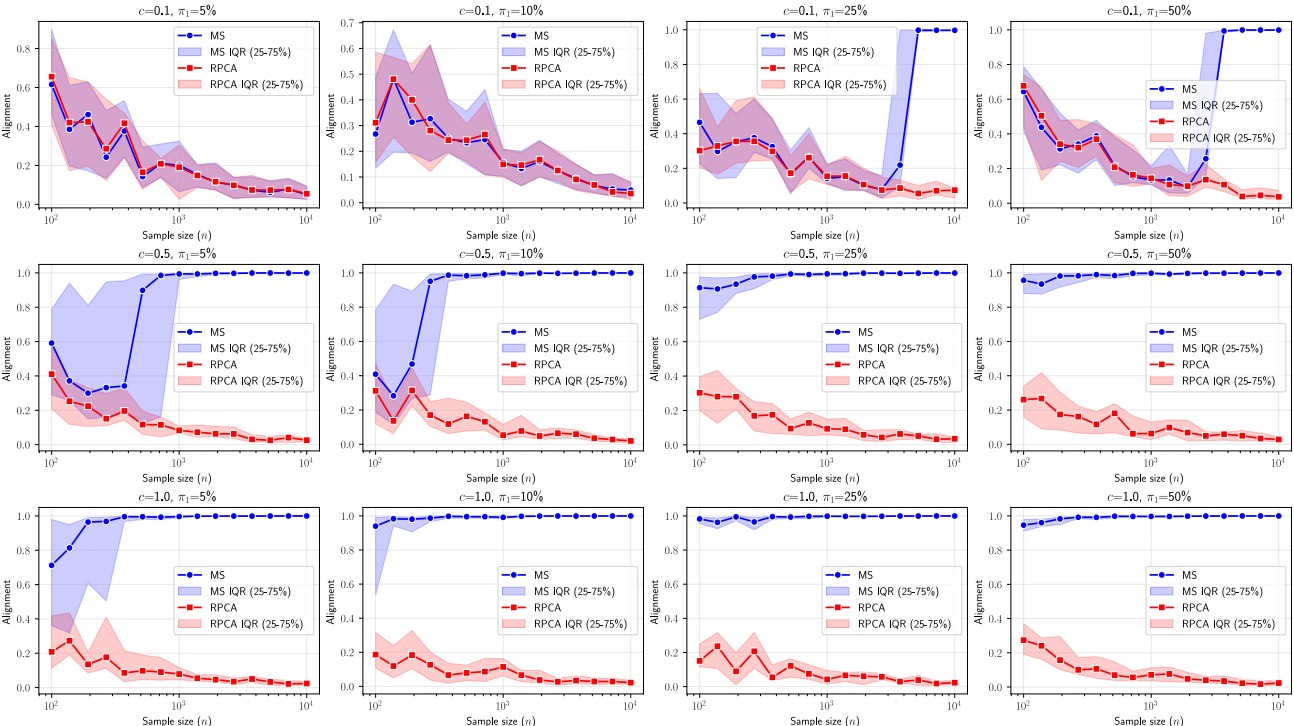

*Figure 6.* **Largest Principal Component Alignment** $|\langle \mathbf{u}_1, \hat{\mathbf{u}}_1 \rangle|$ between the 2 unitary vectors, the estimated largest principal component (PC) $\hat{\mathbf{u}}_1$ and the true largest PC $\mathbf{u}_1$ for MS-PCA and Robust PCA via AAP across dimensions, with varying contamination proportion $\pi_1$ and aspect ratio $c = d/n$ in the Gaussian setting. As the dimension increases, the interquartile range (IQR) shrinks due to concentration of measure. The IQR is computed from 25 independent trials. In the relatively low-dimensional regime ($c = 0.1$), both methods perform comparably poorly when the contamination proportion is not significant. However, our method achieves perfect recovery of the true PC when the contamination proportion is large enough (e.g., $\pi_1 \geq 25\%$) even in this relatively low-dimensional setting. In the high-dimensional regimes, our method consistently outperforms Robust PCA across all contamination proportions.

stable when averaging over many components for large $K$. In this work, we prove that when the contamination happens in the mean direction, instead of in the covariance or higher moments structure, the instability of principal components manifests solely in the relative ordering of the top $K$ contaminated principal components. The true principal components persist within the set of contaminated principal components, albeit with altered ranks. Based on this observation, we introduce a novel recovery algorithm by checking the invariance property of stable principal components. Numerical experiments on spiked Gaussian models demonstrate that our method outperforms the robust PCA in high-dimensional regimes, particularly when the aspect ratio of dimension to sample size is not close to zero.

Furthermore, the term "Robust PCA" is arguably a misnomer—a point subtly acknowledged by the question mark in the title of the original Robust PCA paper by Candès et al. (2011). The prevailing formulation addresses the recovery of a low-rank matrix from sparse, large-magnitude noise, which diverges from the classical definition of robustness in statistics concerning resistance to contamination.

Consequently, a gap exists between the robustly recovered low-rank matrix and the true leading $K$ principal components. While Robust PCA is frequently validated through applications like image background-foreground separation, this task does not align with a traditional statistical robustness problem. This misalignment may explain why Robust PCA is seldom employed in domains where robust principal components are genuinely required, such as population genetics, where standard PCA-based findings can be significantly biased (Elhaik, 2022).

While our theory and method effectively address mean-shift contamination, developing a general solution for arbitrary contamination is an open problem. Our theory is developed for homoscedastic mixtures, prior results exist for zero-mean mixtures with heterogeneous covariances (Benaych-Georges & Couillet, 2016). The broader problem of arbitrary mixtures involving means, covariances, and higher moments remains open and is left for future work.

## Acknowledgements

Zeng Li's research is partially supported by the National Key R&D Program of China (No. 2023YFA1011400) and the National Natural Science Foundation of China (NSFC, No. 12471258). J. Yao's research is supported by NSFC Grant RFIS-A10120230617, the Guangdong Pearl River Talent Program Grant 2023JC10X022, and the Guangdong Provincial Key Laboratory of Mathematical Foundations for Artificial Intelligence (2023B1212010001).

We thank the reviewers for their constructive comments. We acknowledge Junpeng Zhu for early discussions on the matricial formulation. We also thank Francis Bach, Yuda Wu, and Yingyu Yang for reading the draft and providing helpful comments.

## Impact Statement

This paper presents work whose goal is to advance the field of Machine Learning. There are many potential societal consequences of our work, none of which we feel must be specifically highlighted here.

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

## A. Proof of Theorem 3.5 and Lemma 3.8

We prove Theorem 3.5 by first establishing Lemma 3.8, from which the theorem follows directly.

Let $\lambda^{(n)}$ be a spiked eigenvalue of $\widetilde{\mathbf{X}}_n\widetilde{\mathbf{X}}_n^\top/n$ , i.e., $\lambda^{(n)} \notin [(1-\sqrt{c})^2, (1+\sqrt{c})^2] \cup \{0\}$, $\det\left(\widetilde{\mathbf{X}}_n\widetilde{\mathbf{X}}_n^\top/n - \lambda^{(n)}\mathbf{I}_d\right) = 0$.
The resolvent $\mathbf{Q}_n(\lambda^{(n)}) = \left(\mathbf{Z}_n\mathbf{Z}_n^\top/n - \lambda^{(n)}\mathbf{I}_d\right)^{-1}$ is almost surely non-singular for large $n$.

Case 1: If $\lambda^{(n)}$ is also a spiked eigenvalue of $\mathbf{X}_n\mathbf{X}_n^\top/n$, i.e., $\det\left(\mathbf{X}_n\mathbf{X}_n^\top/n - \lambda^{(n)}\mathbf{I}_d\right) = 0$, then $\lambda^{(n)}$ must follow the equation 8 (Baik & Silverstein, 2006; Couillet & Liao, 2022, Theorem 2.13).

Case 2: When $\lambda^{(n)}$ is not an eigenvalue of $\mathbf{X}_n\mathbf{X}_n^\top/n$, i.e., $\det\left(\mathbf{X}_n\mathbf{X}_n^\top/n - \lambda^{(n)}\mathbf{I}_d\right) \neq 0$, we have the following lemma characterizing the spiked eigenvalues of $\widetilde{\mathbf{X}}_n\widetilde{\mathbf{X}}_n^\top/n$ by a finite dimensional matrix:

**Lemma A.1.** *Let* $\lambda^{(n)} \geq 0$ *s.t.* $\det\left(\mathbf{X}_n\mathbf{X}_n^\top/n - \lambda^{(n)}\mathbf{I}_d\right) \neq 0$, *and the matrix* $\mathbf{M}_n \in \mathbb{R}^{2k\times 2k}$ *defined as*

$$\mathbf{M}_n(\lambda^{(n)}) = \begin{pmatrix} \sqrt{\lambda^{(n)}}\mathbf{V}_n^\top(\mathbf{X}_n\mathbf{X}_n^\top/n - \lambda^{(n)}\mathbf{I}_d)^{-1}\mathbf{V}_n & \mathbf{V}_n^\top(\mathbf{X}_n\mathbf{X}_n^\top/n - \lambda^{(n)}\mathbf{I}_d)^{-1}\mathbf{X}_n\mathbf{W}_n/\sqrt{n} \\ \mathbf{W}_n^\top\mathbf{X}_n^\top(\mathbf{X}_n\mathbf{X}_n^\top/n - \lambda^{(n)}\mathbf{I}_d)^{-1}\mathbf{V}_n/\sqrt{n} & \sqrt{\lambda^{(n)}}\mathbf{W}_n^\top(\mathbf{X}_n^\top\mathbf{X}_n/n - \lambda^{(n)}\mathbf{I}_n)^{-1}\mathbf{W}_n \end{pmatrix} - \begin{pmatrix} 0 & \boldsymbol{\Theta}^{-1} \\ \boldsymbol{\Theta}^{-1} & 0 \end{pmatrix},$$

$\det\left(\widetilde{\mathbf{X}}_n\widetilde{\mathbf{X}}_n^\top/n - \lambda^{(n)}\mathbf{I}_d\right) = 0$ *if and only if the matrix* $\mathbf{M}_n(\lambda^{(n)})$ *is singular.*

*Proof.* The proof is essentially given in (Benaych-Georges & Nadakuditi, 2012, Lemma 4.1.) based on the result of (Benaych-Georges et al., 2011, Lemma 6.1.). $\square$

The diagonal blocks concentrate whereas the non-diagonal parts vanish asymptotically. Begin with the diagonal terms:

$$\begin{aligned}
&\mathbf{V}_n^\top\left(\mathbf{X}_n\mathbf{X}_n^\top/n - \lambda^{(n)}\mathbf{I}_d\right)^{-1}\mathbf{V}_n \\
&= \mathbf{V}_n^\top\left((\mathbf{I}_d + \mathbf{P})^{\frac{1}{2}}\mathbf{Z}_n\mathbf{Z}_n^\top(\mathbf{I}_d + \mathbf{P})^{\frac{1}{2}}/n - \lambda^{(n)}\mathbf{I}_d\right)^{-1}\mathbf{V}_n \\
&= \mathbf{V}_n^\top(\mathbf{I}_d + \mathbf{P})^{-\frac{1}{2}}\left(\mathbf{Z}_n\mathbf{Z}_n^\top/n - \lambda^{(n)}\mathbf{I}_d + \lambda^{(n)}(\mathbf{I}_d + \mathbf{P})^{-1}\mathbf{P}\right)^{-1}(\mathbf{I}_d + \mathbf{P})^{-\frac{1}{2}}\mathbf{V}_n && \text{(By Lemma C.4)} \\
&= \mathbf{V}_n^\top(\mathbf{I}_d + \mathbf{P})^{-\frac{1}{2}}\mathbf{Q}_n(\lambda^{(n)})(\mathbf{I}_d + \mathbf{P})^{-\frac{1}{2}}\mathbf{V}_n \\
&\quad - \lambda^{(n)}\mathbf{V}_n^\top(\mathbf{I}_d + \mathbf{P})^{-\frac{1}{2}}\mathbf{Q}_n(\lambda^{(n)})\mathbf{U}_n(\mathbf{I}_r + \mathbf{L}^{-1} + \lambda^{(n)}\mathbf{U}_n^\top\mathbf{Q}_n(\lambda^{(n)})\mathbf{U}_n)^{-1}\mathbf{U}_n^\top\mathbf{Q}_n(\lambda^{(n)})(\mathbf{I}_d + \mathbf{P})^{-\frac{1}{2}}\mathbf{V}_n
\end{aligned}$$

We adopt the following notations in the proof: $\mathbf{P} = \sum_{i=1}^r \ell_i\mathbf{u}_i\mathbf{u}_i^\top = \mathbf{U}_n\mathbf{L}\mathbf{U}_n^\top$ where $\mathbf{U}_n = \left((\mathbf{u}_i)_{i=1}^r\right) \in \mathbb{R}^{d\times r}$, $\mathbf{L} = \text{diag}\left(\ell_i\right)_{i=1}^r$. The deduction is principally based on the proof in (Paul, 2007) and (Couillet & Liao, 2022, Theorem 2.14). For the second equality, we use the resolvent identity Lemma C.4 $(\mathbf{I}_d + \mathbf{P})^{-1} = \mathbf{I}_d - (\mathbf{I}_d + \mathbf{P})^{-1}\mathbf{P}$. The last equality follows from Lemma C.5 $(\mathbf{I}_d + \mathbf{P})^{-1}\mathbf{P} = \mathbf{U}_n(\mathbf{I}_r + \mathbf{L})^{-1}\mathbf{L}\mathbf{U}_n^\top$ and the Woodbury identity Lemma C.6.

As $\mathbf{P}$ and $\mathbf{U}_n$ are finite-rank, $\mathbf{Q}_n(\lambda^{(n)})$ has deterministic equivalent as $S_c(\lambda^{(n)})\mathbf{I}_d$ (Marčenko & Pastur, 1967; Couillet & Liao, 2022, Theorem 2.4) and $\mathbf{Q}_n(\lambda^{(n)})$ is independent of $\mathbf{V}_n$ and $\mathbf{U}_n$, by Proposition C.3,

$$\mathbf{V}_n^\top(\mathbf{I}_d + \mathbf{P})^{-\frac{1}{2}}\mathbf{Q}_n(\lambda^{(n)})(\mathbf{I}_d + \mathbf{P})^{-\frac{1}{2}}\mathbf{V}_n \xrightarrow[a.s.]{n\to\infty} S_c(\lambda^{(n)})\mathbf{I}_k$$

and

$$\mathbf{V}_n^\top(\mathbf{I}_d + \mathbf{P})^{-\frac{1}{2}}\mathbf{Q}_n(\lambda^{(n)})\mathbf{U}_n(\mathbf{I}_r + \mathbf{L}^{-1} + \lambda^{(n)}\mathbf{U}_n^\top\mathbf{Q}_n(\lambda^{(n)})\mathbf{U}_n)^{-1}\mathbf{U}_n^\top\mathbf{Q}_n(\lambda^{(n)})(\mathbf{I}_d + \mathbf{P})^{-\frac{1}{2}}\mathbf{V}_n \xrightarrow[a.s.]{n\to\infty} \mathbf{0}_{k\times k}$$

Then it is well known (Couillet & Liao, 2022, Theorem 2.6) that the resolvent $(\mathbf{X}_n^\top\mathbf{X}_n/n - \lambda^{(n)}\mathbf{I}_n)^{-1} = (\mathbf{Z}_n^\top(\mathbf{I}_d + \mathbf{P})\mathbf{Z}_n/n - \lambda^{(n)}\mathbf{I}_n)^{-1}$ has deterministic equivalent as $S_{1/c}(\lambda^{(n)})\mathbf{I}_n$, so

$$\mathbf{W}_n^\top(\mathbf{X}_n^\top\mathbf{X}_n/n - \lambda^{(n)}\mathbf{I}_n)^{-1}\mathbf{W}_n \xrightarrow[a.s.]{n\to\infty} S_{1/c}(\lambda^{(n)})\mathbf{I}_k.$$

For the non-diagonal terms, the spectral norms of $(\mathbf{X}_n\mathbf{X}_n^\top/n - \lambda^{(n)}\mathbf{I}_d)^{-1}$, $\mathbf{X}_n/\sqrt{n}$ and $\mathbf{W}_n$ are bounded, so we can apply Proposition C.3 to prove

$$\mathbf{V}_n^\top(\mathbf{X}_n\mathbf{X}_n^\top/n - \lambda^{(n)}\mathbf{I}_d)^{-1}\mathbf{X}_n\mathbf{W}_n/\sqrt{n} \xrightarrow[a.s.]{n\to\infty} \mathbf{0}_{k\times k}$$

and

$$\mathbf{W}_n^\top\mathbf{X}_n^\top(\mathbf{X}_n\mathbf{X}_n^\top/n - \lambda^{(n)}\mathbf{I}_d)^{-1}\mathbf{V}_n/\sqrt{n} \xrightarrow[a.s.]{n\to\infty} \mathbf{0}_{k\times k}$$

In summary,

$$\mathbf{M}_n \xrightarrow[a.s.]{n\to\infty} \begin{pmatrix} \sqrt{\lambda^{(n)}}S_c(\lambda^{(n)})\mathbf{I}_k & -\mathbf{\Theta}^{-1} \\ -\mathbf{\Theta}^{-1} & \sqrt{\lambda^{(n)}}S_{1/c}(\lambda^{(n)})\mathbf{I}_k \end{pmatrix}.$$

By the block matrix determinant formula,

$$\det\begin{pmatrix} \sqrt{\lambda^{(n)}}S_c(\lambda^{(n)})\mathbf{I}_k & -\mathbf{\Theta}^{-1} \\ -\mathbf{\Theta}^{-1} & \sqrt{\lambda^{(n)}}S_{1/c}(\lambda^{(n)})\mathbf{I}_k \end{pmatrix} = \prod_{i=1}^k \left(\lambda^{(n)}S_c(\lambda^{(n)})S_{1/c}(\lambda^{(n)}) - \theta_i^{-2}\right),$$

thereby completing the proof.

## B. Proof of Theorem 3.11

Under the setting of Theorem 3.11, we have the following decomposition of the residual vector $\mathbf{r}_n$:

$$\begin{aligned} \mathbf{r}_n &= \frac{1}{n}\left(\widetilde{\mathbf{X}}_n\widetilde{\mathbf{X}}_n^\top - \mathbf{X}_n\mathbf{X}_n^\top\right)\mathbf{u} \\ &= \frac{1}{n}\left(\mathbf{A}\mathbf{A}^\top + \mathbf{A}\mathbf{X}^\top + \mathbf{X}\mathbf{A}^\top\right)\mathbf{u} \end{aligned}$$

Hence

$$\|\mathbf{r}_n\|_2 \leq \frac{1}{n}\left(\sqrt{\mathbf{u}^\top\mathbf{A}\mathbf{A}^\top\mathbf{A}\mathbf{A}^\top\mathbf{u}} + \sqrt{\mathbf{u}^\top\mathbf{X}\mathbf{A}^\top\mathbf{A}\mathbf{X}^\top\mathbf{u}} + \sqrt{\mathbf{u}^\top\mathbf{A}\mathbf{X}^\top\mathbf{X}\mathbf{A}^\top\mathbf{u}}\right)$$

The first term can be bounded as:

$$\frac{1}{n^2}\left|\mathbf{u}^\top\mathbf{A}\mathbf{A}^\top\mathbf{A}\mathbf{A}^\top\mathbf{u}\right| \leq \underbrace{\left\|\frac{\mathbf{A}^\top\mathbf{A}}{n}\right\|_{op}}_{\mathcal{O}(1)}\underbrace{\left\|\frac{\mathbf{A}^\top}{\sqrt{n}}\mathbf{u}\right\|_2^2}_{\mathcal{O}_p(n^{-1})}$$

where the bound $\mathcal{O}_p(n^{-1})$ comes from Proposition C.3 since $\mathbf{u}$ is independent of $\mathbf{A}$ and $\|\mathbf{w}_{(i)}\| = \mathcal{O}(1)$.

In the same way, we can prove $\left\|\frac{\mathbf{A}}{\sqrt{n}}\frac{\mathbf{X}^\top}{\sqrt{n}}\mathbf{u}\right\|_2 = \mathcal{O}_p(n^{-1/2})$ for the second term, which concludes the proof.

## C. Mathematical Tools

**Proposition C.1** (Reverse Mapping for the Largest Eigenvalues). *The spike-forward mapping $g : (\sqrt{c}, \infty) \to ((1+\sqrt{c})^2, \infty)$ defined by*

$$g(\ell) = 1 + \ell + c\frac{1+\ell}{\ell}$$

*is invertible. Its inverse $g^{-1} : ((1+\sqrt{c})^2, \infty) \to (\sqrt{c}, \infty)$ is given explicitly by*

$$g^{-1}(\lambda) = \frac{1}{2}\left(\lambda - 1 - c + \sqrt{(\lambda-1-c)^2 - 4c}\right).$$

*Remark* C.2 (Phase Transition Threshold for General Distribution). We can define the D-transform as $D(\lambda) :=$ $\lambda S_{\mu_\infty}(\lambda)S_{\mu'_\infty}(\lambda)$. For $c \in [0, 1]$ and $\lambda$ outside the support of $\mu_\infty$, it can be written as

$$D_{\mu_\infty}(\lambda) = \left(\int \frac{\lambda}{\lambda - t}d\mu_\infty(t)\right)$$
$$\left(c\int \frac{\lambda}{\lambda - t}d\mu_\infty(t) + \frac{1-c}{\lambda}\right)$$

Let $\lambda^+$ denote the supremum of the support of $\mu_\infty$, the phase transition for contamination-induced largest spectral spikes occurs if $\theta_i^2 > 1/D_{\mu_\infty}(\lambda^+ + \epsilon)$; otherwise, these eigenvalues remain submerged within the bulk spectrum. *The function $D_{\mu_\infty}$ is invertible outside the support of LSD on $\mathbb{R}^+$ and generalizes the reverse mapping g from Proposition C.1 to arbitrary compactly supported distributions* (Benaych-Georges & Nadakuditi, 2012).

**Proposition C.3** (High-dimensional Orthogonality). *Let $\mathbf{u} \in \mathbb{R}^d$ be one column/row vector of a Haar distributed orthogonal matrix, or a random vector with i.i.d. entries with zero mean and vanishing ($\mathcal{O}(d^{-1})$) variance,*

$\mathbf{v} \in \mathbb{R}^d$ *a random (or deterministic) vector with bounded Euclidean norm which is independent of $\mathbf{u}$.*

*Asymptotically when $d \to \infty$, $\mathbf{u}, \mathbf{v}$ are a.s. orthogonal, i.e. $\langle \mathbf{u}, \mathbf{v} \rangle \xrightarrow[a.s.]{d \to \infty} 0$, and $|\langle \mathbf{u}, \mathbf{v} \rangle| = \mathcal{O}(d^{-1/2})$ with high probability.*

*Proof.* Even though similar results can be found in many existing works (Benaych-Georges & Nadakuditi, 2011, Lemma A.2.), we repeat a proof here.

Let $Y_n = \langle \mathbf{u}, \mathbf{v} \rangle = \sum_{i=1}^d u_i v_i$. We will prove that $\mathbb{E}(|Y_n|^2) = \mathcal{O}(d^{-1})$, so $Y_n \xrightarrow[a.s.]{d \to \infty} 0$ by Markov's inequality and $|Y_n| = \mathcal{O}_p(d^{-1/2})$ by Chebyshev's inequality.

When $\mathbf{u}$ is a vector from Haar distributed orthogonal (or unitary) matrix:

$$\mathbb{E}(|Y_n|^2) = \mathbb{E}\left(\sum_{i=1}^d u_i v_i \sum_{j=1}^d u_j v_j\right) = \sum_{i,j=1}^d \mathbb{E}(u_i u_j)\mathbb{E}(v_i v_j) \quad \text{by independence}$$

$$= \sum_{i=1}^d \mathbb{E}(u_i^2)\mathbb{E}(v_i^2) = (d^{-1} + \mathcal{O}(d^{-2})) \underbrace{\sum_{i=1}^d \mathbb{E}(v_i^2)}_{=\mathbb{E}(\sum_{i=1}^d v_i^2)=\mathcal{O}(1)} \quad \text{by (Collins \& Śniady, 2006)}$$

$$= \mathcal{O}(d^{-1}).$$

Based on the results of B. Collins, $\mathbb{E}(u_i u_j) = 0$ if $i \neq j$ (Collins & Śniady, 2006, Corollary 3.4), and $\mathbb{E}(|u_i|^2) = d^{-1} + \mathcal{O}(d^{-2})$ (Collins & Śniady, 2006, Theorem 3.13).

When $\mathbf{u}$ has i.i.d. entries with zero mean and variance $\mathcal{O}(d^{-1})$:

$$\mathbb{E}(|Y_n|^2) = \sum_{i=1}^d \mathbb{E}(|u_i|^2)\mathbb{E}(|v_i|^2) = \mathcal{O}(d^{-1})\underbrace{\sum_{i=1}^d \mathbb{E}(|v_i|^2)}_{=\mathcal{O}(1)} \quad \text{by i.i.d. assumption}$$

$$= \mathcal{O}(d^{-1}).$$

In both cases, we have $\mathbb{E}(|Y_n|^2) = \mathcal{O}(d^{-1})$, which concludes the proof. $\qquad\square$

**Lemma C.4** (Resolvent identity). *For invertible matrices $\mathbf{A}$ and $\mathbf{B}$,*

$$\mathbf{A}^{-1} - \mathbf{B}^{-1} = \mathbf{A}^{-1}(\mathbf{B} - \mathbf{A})\mathbf{B}^{-1}.$$

*Proof.* See (Couillet & Liao, 2022, Lemma 2.1.). $\qquad\square$

**Lemma C.5.** *Let* $\mathbf{A} \in \mathbb{K}^{k \times d}$ *and* $\mathbf{B} \in \mathbb{K}^{d \times k}$,

$$\mathbf{A}(\mathbf{BA} - z\mathbf{I}_d)^{-1} = (\mathbf{AB} - z\mathbf{I}_k)^{-1}\mathbf{A},$$

*for* $z \in \mathbb{C}$ *distinct from* 0 *and from the eigenvalues of* $\mathbf{AB}$.

*Proof.* See (Couillet & Liao, 2022, Lemma 2.2.). □

With Lemma C.5, we can deduce a useful equation to shift the dimension: $(\mathbf{I}_d + \mathbf{P})^{-1}\mathbf{U}_n = (\mathbf{I}_d + \mathbf{U}_n\mathbf{LU}_n^\intercal)^{-1}\mathbf{U}_n = \mathbf{U}_n(\mathbf{I}_r + \mathbf{LU}_n^\intercal\mathbf{U}_n)^{-1} = \mathbf{U}_n(\mathbf{I}_r + \mathbf{L})^{-1}$.

**Lemma C.6** (Woodbury matrix identity)**.** *Let* $\mathbf{A}$, $\mathbf{U}$, $\mathbf{C}$, *and* $\mathbf{V}$ *be conformable matrices, where* $\mathbf{A}$ *is* $n \times n$, $\mathbf{C}$ *is* $k \times k$, $\mathbf{U}$ *is* $n \times k$, *and* $\mathbf{V}$ *is* $k \times n$. *Then:*

$$(\mathbf{A} + \mathbf{UCV})^{-1} = \mathbf{A}^{-1} - \mathbf{A}^{-1}\mathbf{U}(\mathbf{C}^{-1} + \mathbf{VA}^{-1}\mathbf{U})^{-1}\mathbf{VA}^{-1},$$

*Proof.* This identity can be derived using blockwise matrix inversion, or by checking directly $(\mathbf{A} + \mathbf{UCV})(\mathbf{A}^{-1} - \mathbf{A}^{-1}\mathbf{U}(\mathbf{C}^{-1} + \mathbf{VA}^{-1}\mathbf{U})^{-1}\mathbf{VA}^{-1}) = \mathbf{I}_n$. □

# D. Related Work in Random Matrix Theory

**Additive Low-Rank Random Perturbation** The statistical behavior of low-rank matrix perturbations has been extensively studied within random matrix theory. Seminal work by Benaych-Georges and colleagues (Benaych-Georges et al., 2011; Benaych-Georges & Nadakuditi, 2012) established a comprehensive framework for random matrix under additive low-rank perturbation, assuming the limiting spectral distribution of the unperturbed matrix has compact support. Their analysis derived exact phase transitions for singular values/vectors of the perturbed matrix under high-dimensional regimes, including CLT-scale fluctuations.

Especially, they demonstrated that when the singular values of the perturbation exceed a critical threshold, the corresponding singular values of the perturbed matrix separate from the bulk spectrum, and the associated singular vectors exhibit non-trivial alignment with those of the perturbation. Existing results necessitate strong conditions on the alignment of $\mathbf{A}_n$'s both left and right singular vectors— requiring either:

- i.i.d. zero mean and unit variance entries satisfying a log-Sobolev inequality, or

- uniformly random unit vector as Haar-distributed orthonormal columns

This condition on the right singular vectors does not hold for our case, because our indicator vectors $\boldsymbol{\gamma}_i \in \{0,1\}^n$ exhibit a special binary dependent orthogonal structure. We can extend these results to our mean-shift setting (1).

**Multiplicative Low-Rank Perturbation in i.i.d. Model** Complementary to additive low-rank perturbation models, seminal work on spiked population covariance matrices (Baik & Silverstein, 2006; Paul, 2007) analyzes the spectral behavior of random matrix under multiplicative low-rank perturbations:

$$(\mathbf{I}_d + \mathbf{P})^{\frac{1}{2}}\mathbf{Z}_n,$$

where $\mathbf{Z}_n \in \mathbb{R}^{d \times n}$ has i.i.d. entries with zero mean, unit variance, and finite fourth moment. $\mathbf{P}$ is a finite-rank matrix, and $(\mathbf{I}_d + \mathbf{P})^{1/2}$ induces population covariance. These studies establish fundamental phase transitions for the extreme eigenvalues and corresponding eigenvectors of the sample covariance matrix.

**Higher Rank Perturbation in i.i.d. Model** Recent work by (Liu et al., 2025) analyzes the perturbed population model $\mathbf{A}_n + \boldsymbol{\Sigma}^{1/2}\mathbf{Z}_n$, when the rank of $\mathbf{A}_n$ scales as $\mathcal{O}(n^{1/3})$. While they derive asymptotic limits for spiked eigenvalues/eigenvectors, their results require additional assumption involving integral of the empirical spectral distribution of the latent matrix

$$\mathbf{R}_n := \mathbf{A}_n\mathbf{A}_n^\top + \boldsymbol{\Sigma},$$

which is difficult to verify in practice. Their eigenspace estimation results for the sample covariance matrix depend on complete spectral knowledge of the unobservable latent matrix $\mathbf{R}_n$ and the resulting estimators are not interpretable for application problems.

*Table 2.* **Comparison with robust estimators and preprocessing baselines.** Alignment of the largest principal component under mean-shift Gaussian mixture contamination. Values are percentages, reported as mean $\pm$ standard deviation over 200 independent trials. Here $d = 900$, $n = 10^3$, and higher is better.

| $\pi_1$ | MS-PCA | RPCA-AAP | Tyler | Huber | $\ell_1$-PCA | winsorized-PCA | center-PCA |
|---|---|---|---|---|---|---|---|
| 5% | $95.85 \pm 12.53$ | $8.40 \pm 6.87$ | $9.33 \pm 7.46$ | $9.72 \pm 7.84$ | $14.01 \pm 10.73$ | $9.26 \pm 7.42$ | $9.27 \pm 7.37$ |
| 10% | $97.16 \pm 6.96$ | $8.75 \pm 6.71$ | $10.67 \pm 8.00$ | $10.95 \pm 8.11$ | $14.25 \pm 10.46$ | $10.63 \pm 8.01$ | $10.64 \pm 8.10$ |
| 15% | $97.39 \pm 7.03$ | $7.47 \pm 6.13$ | $10.24 \pm 8.12$ | $10.51 \pm 8.22$ | $12.06 \pm 9.33$ | $10.22 \pm 8.09$ | $10.36 \pm 8.07$ |
| 20% | $96.17 \pm 11.82$ | $7.74 \pm 5.67$ | $11.46 \pm 8.63$ | $11.53 \pm 8.72$ | $11.85 \pm 8.81$ | $11.47 \pm 8.63$ | $11.70 \pm 8.70$ |

*Table 3.* **Runtime for estimating leading PCs.** Runtime in milliseconds for $d = 900$, $n = 1000$, reported as mean $\pm$ standard deviation.

| | MS-PCA | RPCA-AAP | Tyler | Huber | $\ell_1$-PCA |
|---|---|---|---|---|---|
| 1 PC | $14.2 \pm 0.0137$ | $79.0 \pm 0.176$ | $172 \pm 0.159$ | $3860 \pm 4.37$ | $8230 \pm 294$ |
| 9 PCs | $19.2 \pm 0.131$ | $99.6 \pm 0.210$ | $174 \pm 0.351$ | $3860 \pm 8.5$ | N/A |

# E. Experiment and Implementation Detail

Experiments were performed on the AMD EPYC 9755 256-core server (dual-socket with 128 cores per socket) without GPU acceleration. All Gaussian experiments follow the same parameter configuration detailed in Section 4. We release an implementation at `https://github.com/Mengda-Li/ms-pca`.

**Runtime comparison** We report empirical runtimes for the comparison experiments in Table 2. Timing is measured by `%timeit`; all values are in milliseconds and are reported as mean $\pm$ standard deviation. Table 3 compares the cost of estimating one and nine leading principal components for MS-PCA, RPCA-AAP, Tyler's M estimator, Huber's M estimator, and $\ell_1$-PCA. Table 4 compares MS-PCA and RPCA-AAP as the dimension grows with $c = d/n = 1$. MS-PCA is more than five times faster than RPCA-AAP in the base setting, and the runtime ratio increases from roughly 5.4 to 12.6 as $d$ grows from 1000 to 10000.

**Mean-shift plus covariance-shift contamination** To test a more complex corruption pattern, we also consider a mixture with both mean-shift and covariance-shift contamination:

$$\mathcal{N}(0, \mathbf{I}_d + \mathbf{P}_1) \quad \text{and} \quad \mathcal{N}(\mathbf{m}_1, (\mathbf{I}_d + \mathbf{P}_1)(\mathbf{I}_d + \mathbf{P}_2)).$$

Here $\pi_1$ is the weight of the second component and $\ell_2$ is the spike of the rank-one covariance perturbation $\mathbf{P}_2$. Table 5 reports recovery of the largest principal component over 100 independent trials. MS-PCA remains above 95% alignment throughout the tested settings, while the competing methods remain below 17%.

*Table 4.* **Runtime scaling with dimension.** Runtime in milliseconds for estimating the largest PC with $c = d/n = 1$, reported as mean $\pm$ standard deviation.

| Method | $d = 1000$ | $d = 2000$ | $d = 3000$ | $d = 4000$ | $d = 5000$ | $d = 10000$ |
|---|---|---|---|---|---|---|
| MS-PCA | $15.9 \pm 0.0109$ | $49.2 \pm 0.0795$ | $102 \pm 0.22$ | $216 \pm 1.1$ | $237 \pm 0.516$ | $832 \pm 1.26$ |
| RPCA-AAP | $86 \pm 0.102$ | $369 \pm 0.945$ | $1050 \pm 3.01$ | $2490 \pm 4.39$ | $2750 \pm 6.78$ | $10500 \pm 21.6$ |

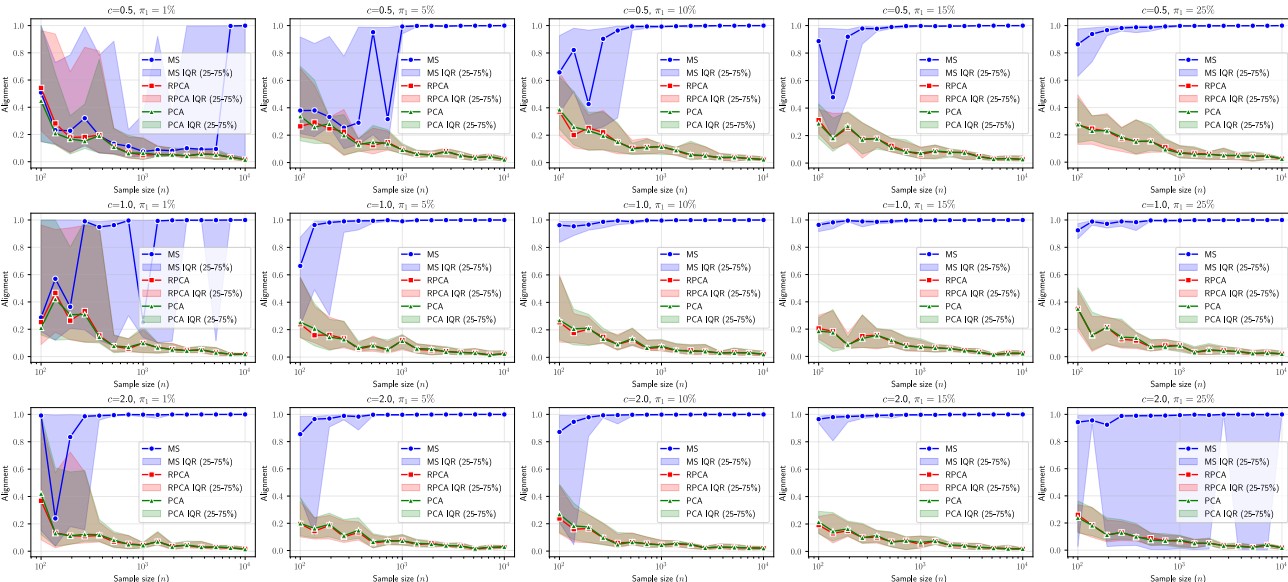

*Figure 7.* Additional Experiment. Alignment $|\langle \mathbf{u}_1, \hat{\mathbf{u}}_1 \rangle|$ of the largest principal component between the 2 unitary vectors, the estimated principal component $\hat{\mathbf{u}}_1$ and the true principal component $\mathbf{u}_1$ for MS-PCA, Robust PCA via AAP and vanilla PCA across dimensions, with varying contamination proportion $\pi_1$ and aspect ratio $c = d/n$ in the Gaussian setting.

*Table 5.* **Mean-shift plus covariance-shift contamination.** Alignment of the largest principal component in percent, reported as mean $\pm$ standard deviation over 100 independent trials. Higher is better.

| $(\pi_1, \ell_2)$ | MS-PCA | RPCA-AAP | Tyler | Huber | $\ell_1$-PCA | winsorized-PCA | center-PCA |
|---|---|---|---|---|---|---|---|
| $(5\%, 2)$ | $97.01 \pm 8.01$ | $7.69 \pm 6.42$ | $8.67 \pm 7.26$ | $9.14 \pm 7.49$ | $13.13 \pm 10.61$ | $8.59 \pm 7.20$ | $8.64 \pm 7.32$ |
| $(5\%, 4)$ | $97.52 \pm 1.76$ | $9.09 \pm 7.90$ | $9.82 \pm 8.20$ | $9.83 \pm 8.37$ | $15.17 \pm 11.83$ | $9.85 \pm 8.19$ | $9.83 \pm 8.14$ |
| $(5\%, 8)$ | $97.15 \pm 8.37$ | $7.84 \pm 5.91$ | $8.58 \pm 6.64$ | $9.06 \pm 6.70$ | $13.46 \pm 10.19$ | $8.50 \pm 6.68$ | $8.54 \pm 6.72$ |
| $(5\%, 16)$ | $96.54 \pm 8.67$ | $9.74 \pm 10.52$ | $10.88 \pm 11.64$ | $11.35 \pm 11.92$ | $16.01 \pm 14.33$ | $10.79 \pm 11.56$ | $10.71 \pm 11.35$ |
| $(10\%, 2)$ | $97.52 \pm 1.45$ | $8.61 \pm 6.21$ | $10.29 \pm 7.37$ | $10.38 \pm 7.53$ | $13.32 \pm 10.05$ | $10.26 \pm 7.36$ | $10.26 \pm 7.36$ |
| $(10\%, 4)$ | $97.80 \pm 1.40$ | $7.42 \pm 6.46$ | $8.81 \pm 7.45$ | $9.16 \pm 7.54$ | $11.60 \pm 9.61$ | $8.74 \pm 7.45$ | $8.78 \pm 7.56$ |
| $(10\%, 8)$ | $97.64 \pm 1.60$ | $8.66 \pm 6.04$ | $10.09 \pm 7.56$ | $10.13 \pm 7.75$ | $13.55 \pm 10.04$ | $10.08 \pm 7.54$ | $10.04 \pm 7.45$ |
| $(10\%, 16)$ | $97.75 \pm 1.27$ | $8.06 \pm 5.77$ | $9.87 \pm 6.82$ | $10.02 \pm 6.97$ | $13.10 \pm 9.07$ | $9.83 \pm 6.82$ | $9.93 \pm 6.73$ |
| $(15\%, 2)$ | $96.63 \pm 9.72$ | $8.33 \pm 6.15$ | $10.74 \pm 7.71$ | $10.97 \pm 7.75$ | $12.20 \pm 9.01$ | $10.70 \pm 7.71$ | $10.79 \pm 7.62$ |
| $(15\%, 4)$ | $97.65 \pm 1.48$ | $7.90 \pm 6.09$ | $9.95 \pm 7.56$ | $9.98 \pm 7.60$ | $11.79 \pm 8.60$ | $9.95 \pm 7.58$ | $10.01 \pm 7.49$ |
| $(15\%, 8)$ | $97.45 \pm 1.29$ | $8.57 \pm 6.22$ | $10.81 \pm 7.77$ | $10.83 \pm 8.06$ | $12.51 \pm 9.10$ | $10.81 \pm 7.74$ | $10.98 \pm 7.77$ |
| $(15\%, 16)$ | $95.68 \pm 13.68$ | $8.97 \pm 6.83$ | $11.82 \pm 8.95$ | $11.93 \pm 8.98$ | $13.45 \pm 10.63$ | $11.79 \pm 8.95$ | $11.71 \pm 9.10$ |
| $(20\%, 2)$ | $97.91 \pm 1.11$ | $7.63 \pm 6.01$ | $11.18 \pm 8.22$ | $11.51 \pm 7.93$ | $11.17 \pm 8.49$ | $11.13 \pm 8.29$ | $11.21 \pm 8.16$ |
| $(20\%, 4)$ | $95.02 \pm 16.58$ | $8.64 \pm 6.41$ | $12.84 \pm 9.51$ | $12.74 \pm 10.07$ | $13.35 \pm 9.90$ | $12.89 \pm 9.40$ | $13.02 \pm 9.28$ |
| $(20\%, 8)$ | $96.76 \pm 9.82$ | $7.93 \pm 6.04$ | $11.33 \pm 9.16$ | $11.81 \pm 9.34$ | $11.63 \pm 9.30$ | $11.27 \pm 9.15$ | $11.41 \pm 9.22$ |
| $(20\%, 16)$ | $96.68 \pm 9.74$ | $7.84 \pm 5.91$ | $11.23 \pm 8.57$ | $11.24 \pm 8.60$ | $11.50 \pm 8.63$ | $11.24 \pm 8.58$ | $11.09 \pm 8.37$ |

*Table 6.* Maximal residual norm $\|\frac{1}{n}\mathbf{X}_n\mathbf{X}_n^\top\tilde{\mathbf{u}} - \tilde{\lambda}\tilde{\mathbf{u}}\|_2$ for eigenvectors associated with stable eigenvalues (for $\tilde{\lambda}$ not in the neighborhood of $\Lambda_\mathbf{A}$) on Gaussian data. Values are presented as max $\times\ 10^{\text{exponent}}$ with two decimal places. $n = d$, $c = 1$.

| MIXTURE WEIGHT $\pi_1$ | $d = 100$ | $d = 1000$ | $d = 10000$ | $d = 50000$ |
|---|---|---|---|---|
| 0.1% | $5.04 \times 10^{-15}$ | $1.50 \times 10^{-1}$ | $5.20 \times 10^{-2}$ | $2.37 \times 10^{-2}$ |
| 1% | $5.04 \times 10^{-15}$ | $1.49 \times 10^{-1}$ | $5.48 \times 10^{-2}$ | $2.88 \times 10^{-2}$ |
| 10% | $3.51 \times 10^{-1}$ | $1.50 \times 10^{-1}$ | $4.84 \times 10^{-2}$ | $2.50 \times 10^{-2}$ |
| 30% | $3.99 \times 10^{-1}$ | $1.57 \times 10^{-1}$ | $4.88 \times 10^{-2}$ | $2.55 \times 10^{-2}$ |
| 50% | $3.55 \times 10^{-1}$ | $1.60 \times 10^{-1}$ | $5.30 \times 10^{-2}$ | $2.58 \times 10^{-2}$ |
| 100% | $3.27 \times 10^{-1}$ | $1.32 \times 10^{-1}$ | $5.26 \times 10^{-2}$ | $2.75 \times 10^{-2}$ |

*Table 7.* Maximal residual norm of eigenvector alignment $\|\frac{1}{n}\widetilde{\mathbf{X}}_n\widetilde{\mathbf{X}}_n^\top\tilde{\mathbf{u}} - \tilde{\lambda}\tilde{\mathbf{u}}\|_2$ for eigenvectors associated with stable $\tilde{\lambda}$s (for $\tilde{\lambda}$ not in the neighborhood of $\Lambda_\mathbf{A}$) on Gaussian data with varying mixture weights and dimensions. Values are presented as max $\times\ 10^{\text{exponent}}$ with one decimal place. $n = d$, $c = 1$.

| MIXTURE WEIGHT $\pi_1$ | $d = 100$ | $d = 1000$ | $d = 10000$ | $d = 50000$ |
|---|---|---|---|---|
| 0.1% | $5.0 \times 10^{-15}$ | $1.5 \times 10^{-1}$ | $5.2 \times 10^{-2}$ | $2.4 \times 10^{-2}$ |
| 1% | $5.0 \times 10^{-15}$ | $1.5 \times 10^{-1}$ | $5.5 \times 10^{-2}$ | $2.9 \times 10^{-2}$ |
| 10% | $3.5 \times 10^{-1}$ | $1.5 \times 10^{-1}$ | $4.8 \times 10^{-2}$ | $2.5 \times 10^{-2}$ |
| 30% | $4.0 \times 10^{-1}$ | $1.6 \times 10^{-1}$ | $4.9 \times 10^{-2}$ | $2.5 \times 10^{-2}$ |
| 50% | $3.6 \times 10^{-1}$ | $1.6 \times 10^{-1}$ | $5.3 \times 10^{-2}$ | $2.6 \times 10^{-2}$ |
| 100% | $3.3 \times 10^{-1}$ | $1.3 \times 10^{-1}$ | $5.3 \times 10^{-2}$ | $2.7 \times 10^{-2}$ |

*Table 8.* Mean and standard deviation of eigenvector alignment $\|\frac{1}{n}\widetilde{\mathbf{X}}_n\widetilde{\mathbf{X}}_n^\top\mathbf{u} - \lambda\mathbf{u}\|_2$ on Gaussian data with varying mixture weights and dimensions. Values are presented as (mean $\pm$ std) $\times\ 10^{\text{exponent}}$ with one decimal place. $n = d$, $c = 1$.

| MIXTURE WEIGHT $\pi_1$ | $d = 100$ | $d = 1000$ | $d = 10000$ | $d = 50000$ |
|---|---|---|---|---|
| 0.1% | $(3.6 \pm 0.6) \times 10^{-15}$ | $(1.3 \pm 0.9) \times 10^{-1}$ | $(3.4 \pm 2.3) \times 10^{-2}$ | $(1.9 \pm 1.4) \times 10^{-2}$ |
| 1.0% | $(3.6 \pm 0.6) \times 10^{-15}$ | $(1.4 \pm 1.0) \times 10^{-1}$ | $(4.0 \pm 2.8) \times 10^{-2}$ | $(1.9 \pm 1.3) \times 10^{-2}$ |
| 10% | $(2.9 \pm 2.2) \times 10^{-1}$ | $(1.4 \pm 0.9) \times 10^{-1}$ | $(4.0 \pm 2.8) \times 10^{-2}$ | $(1.8 \pm 1.3) \times 10^{-2}$ |
| 30% | $(2.6 \pm 1.6) \times 10^{-1}$ | $(1.3 \pm 0.9) \times 10^{-1}$ | $(4.0 \pm 2.9) \times 10^{-2}$ | $(1.8 \pm 1.3) \times 10^{-2}$ |
| 50% | $(4.4 \pm 3.3) \times 10^{-1}$ | $(1.2 \pm 0.9) \times 10^{-1}$ | $(4.0 \pm 2.8) \times 10^{-2}$ | $(1.8 \pm 1.3) \times 10^{-2}$ |
| 100% | $(4.1 \pm 2.8) \times 10^{-1}$ | $(1.2 \pm 0.9) \times 10^{-1}$ | $(4.0 \pm 2.8) \times 10^{-2}$ | $(1.8 \pm 1.3) \times 10^{-2}$ |

*Table 9.* Mean and standard deviation of eigenvector alignment $\|\frac{1}{n}\widetilde{\mathbf{X}}_n\widetilde{\mathbf{X}}_n^\top\tilde{\mathbf{u}} - \lambda\tilde{\mathbf{u}}\|_2$ on Gaussian data with varying mixture weights and dimensions. Values are presented as (mean $\pm$ std) $\times\ 10^{\text{exponent}}$ with one decimal place. $n = d$, $c = 1$.

| MIXTURE WEIGHT $\pi_1$ | $d = 100$ | $d = 1000$ | $d = 10000$ | $d = 50000$ |
|---|---|---|---|---|
| 0.1% | $(4.3 \pm 1.2) \times 10^{-15}$ | $(3.8 \pm 0.3) \times 10^{-15}$ | $(1.8 \pm 1.2) \times 10^{-2}$ | $(7.2 \pm 4.7) \times 10^{-3}$ |
| 1.0% | $(1.6 \pm 1.0) \times 10^{-1}$ | $(6.4 \pm 4.6) \times 10^{-2}$ | $(1.6 \pm 1.1) \times 10^{-2}$ | $(6.3 \pm 4.1) \times 10^{-3}$ |
| 10.0% | $(3.4 \pm 2.5) \times 10^{-1}$ | $(4.9 \pm 3.1) \times 10^{-2}$ | $(1.4 \pm 0.9) \times 10^{-2}$ | $(6.7 \pm 4.4) \times 10^{-3}$ |
| 30.0% | $(2.1 \pm 1.3) \times 10^{-1}$ | $(5.2 \pm 3.7) \times 10^{-2}$ | $(1.6 \pm 1.0) \times 10^{-2}$ | $(7.0 \pm 4.6) \times 10^{-3}$ |
| 50.0% | $(2.0 \pm 1.3) \times 10^{-1}$ | $(5.5 \pm 3.5) \times 10^{-2}$ | $(1.5 \pm 1.0) \times 10^{-2}$ | $(6.8 \pm 4.4) \times 10^{-3}$ |
| 100.0% | $(2.0 \pm 1.3) \times 10^{-1}$ | $(5.1 \pm 3.4) \times 10^{-2}$ | $(1.6 \pm 1.0) \times 10^{-2}$ | $(6.9 \pm 4.5) \times 10^{-3}$ |

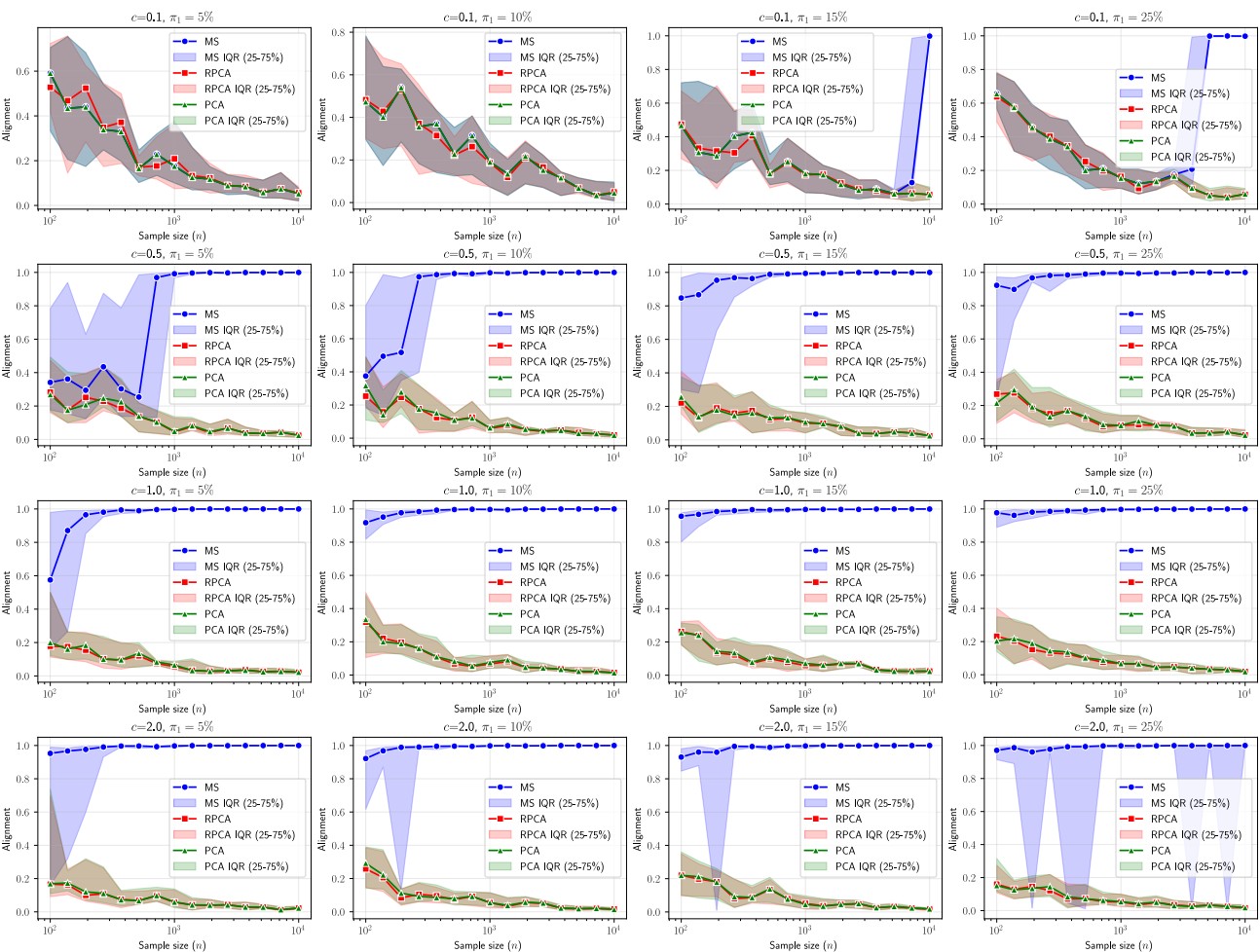

*Figure 8.* Additional Experiment. Alignment $|\langle \mathbf{u}_1, \hat{\mathbf{u}}_1 \rangle|$ of the largest principal component between the 2 unitary vectors, the estimated principal component $\hat{\mathbf{u}}_1$ and the true principal component $\mathbf{u}_1$ for MS-PCA, Robust PCA via AAP and vanilla PCA across dimensions, with varying contamination proportion $\pi_1$ and aspect ratio $c = d/n$ in the Gaussian setting.

