# OpenReview forum: "Mean-Shift PCA by Knockoff Mean"
_ICML.cc/2026/Conference — ICML 2026 regular_

### Official Review · Reviewer_ByR5 · 2026-02-22

**Soundness:** 3
**Presentation:** 3
**Significance:** 2
**Originality:** 3
**Overall Recommendation:** 4
**Confidence:** 3

**Summary:**

# Summary

This manuscript discusses an important aspect of high-dimensional statistical learning: the instability of classical PCA under structured mean-shift contamination. The authors focus on a contamination model in which observations arise from a mixture of a primary distribution (with covariance structure of interest) and one or more subpopulations that differ only in their means. Unlike classical “outlier” formulations where contamination is sparse or arbitrarily corrupted, here the contamination is low-rank in structure but not sparse. This distinction is central to the paper’s positioning.

The authors argue that existing Robust PCA (RPCA) approaches—particularly those based on low-rank plus sparse decompositions—are fundamentally mismatched to this setting because mean-shift contamination is not sparse and may itself exhibit low-rank structure similar to the signal. As a result, RPCA methods do not necessarily recover the principal components of the uncontaminated covariance matrix in high-dimensional regimes.

The core theoretical contribution is an asymptotic spectral decoupling result derived using tools from random matrix theory (RMT). Under spiked covariance and more general compactly supported spectral models, the paper proves that: (1) eigenvalue spikes induced by covariance structure and those induced by mean-shift contamination evolve independently in the large-dimensional limit; and (2) eigenvectors associated with the uncontaminated covariance remain asymptotically invariant under mean-shift contamination up to (O(n^{-1/2})) perturbations.

This spectral independence phenomenon is leveraged algorithmically. The authors propose a two-stage PCA procedure (“MS-PCA”) that intentionally injects an artificial knockoff mean-shift perturbation into the data. Since mean-induced spikes shift under additional mean perturbation while covariance-induced spikes remain invariant, one can identify stable eigenvalues by comparing spectra before and after injection. Principal components corresponding to invariant eigenvalues are retained; unstable ones are removed.

The method uses only standard PCA operations and does not require convex optimization or specialized robust estimators. Empirical experiments on synthetic Gaussian mixture models demonstrate that the proposed method recovers the leading principal component in regimes where RPCA fails, particularly when the dimension-to-sample ratio is not small.

This study's central contribution comprises a unification of RMT-based spectral phase transition analysis with a perturbation-based algorithmic strategy for distinguishing contamination-induced components from intrinsic covariance components. The work is mathematically sophisticated and conceptually elegant, but empirical validation is limited to synthetic settings and idealized assumptions.

**Compliance With Llm Reviewing Policy:**

Affirmed.

**Key Questions For Authors:**

1. **Below-Threshold Contamination:**
   How does MS-PCA behave when mean-shift strength is below the BBP phase transition? Can contamination still be detected?

2. **Sensitivity Analysis:**
   How sensitive is performance to the choice of perturbation magnitude ( \theta' )? Is there a data-driven method for selecting it?

3. **Real Data Performance:**
   Have you evaluated MS-PCA on real-world datasets exhibiting batch effects or mean shifts?

4. **Multiple Spikes:**
   How does the method perform when multiple covariance spikes and multiple contamination spikes are close in magnitude?

5. **Finite Sample Behavior:**
   Can you provide non-asymptotic error bounds or concentration guarantees?

6. **Correlated Contamination:**
   What happens if mean-shift direction aligns partially with true signal eigenvectors?

7. **Comparison to Robust Covariance Estimators:**
   How does MS-PCA compare to Tyler’s estimator or Huberized covariance PCA?

**Limitations:**

The primary limitation is the narrow contamination model. The theory assumes finite-rank mean shifts independent of intrinsic covariance structure. In real applications, contamination may involve covariance heterogeneity, heavy tails, structured dependence, or adversarial alignment.

The guarantees are asymptotic and rely on high-dimensional regimes with fixed aspect ratio. Moderate-dimensional behavior may differ.

The empirical validation is limited to synthetic Gaussian settings, leaving open questions about robustness to non-Gaussian distributions and model misspecification.

Parameter selection remains heuristic and may impact stability.

The method does not address arbitrary contamination, only first-moment shifts. Broader robust PCA problems remain open.

**Strengths And Weaknesses:**

## Strengths

### 1. Clear problem formulation and motivation

The paper identifies a precise structural vulnerability of PCA: sensitivity to first-moment contamination in mixture models. Unlike adversarial corruption or sparse outliers, mean-shift contamination is structured and potentially low-rank, making it particularly insidious in high dimensions. The authors clearly articulate why existing RPCA formulations are not designed for this contamination regime.

The argument that RPCA addresses sparse corruption of a low-rank matrix rather than contamination of covariance structure is conceptually sharp. This positioning is persuasive and grounded in the statistical definition of robustness.

### 2. Strong theoretical backbone

The theoretical analysis is grounded in established RMT literature, including spiked covariance models, BBP phase transitions, and finite-rank perturbation theory. The main eigenvalue characterization theorem (Theorem 3.5) cleanly separates the top eigenvalues into those determined by population covariance spikes and those determined by mean-shift spikes.

The independence between spike families is nontrivial and important. It allows the authors to manipulate one family (mean-induced spikes) without disturbing the other (covariance-induced spikes). This decoupling is conceptually elegant.

The eigenspace invariance theorem (Theorem 3.11) is especially important. It formalizes the idea that uncontaminated eigenvectors remain approximate eigenvectors of the contaminated covariance matrix, up to (O(n^{-1/2})) perturbations. This result provides the key justification for the perturbation-based algorithm.

The proofs appear technically sound and rely appropriately on resolvent methods, deterministic equivalents, and known asymptotic results.

### 3. Algorithmic elegance

The MS-PCA algorithm is conceptually clever:

* Instead of removing contamination directly, inject additional structured noise.
* Use differential spectral response to distinguish contamination.

This “add noise to remove noise” philosophy is unconventional but elegant. Importantly, the algorithm uses only vanilla PCA computations. This makes it computationally lightweight relative to convex RPCA or iterative reweighting methods.

### 4. Conceptual novelty

The key novelty lies not in developing new RMT machinery, but in synthesizing known results into a practical perturbation-based identification strategy. The idea of exploiting spectral phase transition independence for contamination detection is original and intellectually satisfying.

### 5. Clarity of theoretical statements

The theoretical results are clearly stated with explicit assumptions. Phase transition thresholds are derived and generalized via D-transform arguments. The authors carefully delineate when spikes emerge outside the bulk.

---

## Weaknesses

### 1. Strong and restrictive assumptions

The model assumes:

* Finite-rank mean-shift contamination
* Independence between contamination directions and signal
* Homoscedastic mixture components
* Compactly supported limiting spectral distributions
* High-dimensional asymptotics with fixed aspect ratio

In practice, real-world contamination often includes covariance shifts, heavy tails, dependence structures, or correlated subpopulations. The independence assumption between mean-shift directions and intrinsic covariance eigenvectors may fail in real data.

The theory is precise but narrow.

### 2. Limited empirical validation

The experiments are confined to synthetic Gaussian mixtures with one covariance spike. This is insufficient for ICML standards.

Missing evaluations include:

* Real-world datasets (e.g., genomics, embeddings, vision)
* Multiple covariance spikes
* Subspace recovery metrics beyond leading PC alignment
* Sensitivity to parameter choices
* Performance when contamination strength is near or below phase transition
* Runtime comparisons

The claim that RPCA fails in high dimensions is demonstrated only for a specific implementation (AAP). No comparison to robust covariance estimators, Tyler-based PCA, or Huber covariance methods is provided.

### 3. Parameter sensitivity

The method requires selecting:

* Perturbation magnitude ( \theta' )
* Threshold constant ( C )
* Eigenvalue matching tolerance

The paper provides heuristic guidance but no systematic sensitivity analysis. It is unclear how robust the method is to mis-specification.

### 4. Finite-sample guarantees

All guarantees are asymptotic. No non-asymptotic bounds are provided. In practical moderate-dimensional settings, the behavior may differ from asymptotic predictions.

### 5. Limited scope of contamination model

The method addresses only first-moment contamination. The authors acknowledge this, but the limitation is substantial. Real contamination often involves shifts in both mean and covariance.

### 6. Practical detectability below phase transition

If contamination strength is below the BBP threshold, spikes remain inside the bulk. The paper does not analyze detectability in this regime. It is unclear whether MS-PCA offers advantages when spikes are not spectrally separated.

### 7. Risk of overclaiming

The rhetorical positioning that RPCA is fundamentally mismatched may be somewhat overstated without broader empirical comparison.

---

> ### Author Rebuttal · Authors · 2026-03-30
>
> We thank Reviewer ByR5 for the thorough and balanced review. We address the key questions and concerns below.
>
> ## Answers to Questions
>
> **1. Below‑Threshold Contamination**
>
> When the mean‑shift strength of the original noise \\(A\\) is below the BBP phase transition: the contamination noise has no spectral impact, and MS-PCA becomes neutral. Therefore, contamination is too weak to be detected  and covariance spikes can be detected by any standard PCA methods.
>
>
> **2. Sensitivity Analysis for the Choice of \\(\\theta'\\)**
>
> The performance of our algorithm is indeed sensitive to the choice of \\(\\theta'\\).  If \\(\\theta'\\) is too small, no new spectral spike is generated.  We recommend choosing \\(\\theta'\\) such that the newly generated spike(s) are of the same order as the existing spikes. In the Gaussian mixture case, one can use a closed‑form formula to compute the desired \\(\\theta'\\) via the inverse of the spike‑forward mapping \\(g\\) (the equation at line 161, right column, or the \\(D\\)-transform for the general case) to the currently observed spectral spike eigenvalue(s). Details can be found in Lines 156–164 (right column) in the paper.
>
> **3. Real Data Performance**
>
> If the contamination in real data follows a mean‑shift pattern, our algorithm can address the problem with theoretical guarantees (under compact spectral support). However, real‑world noise may follow more complex patterns that can not be captured by mean-shift solely. That's why the paper doesn't contain real data analysis at the moment. We are continuing to extend our modelling to catch up with real data examples.
>
> **4. Multiple Spikes Which Are Close**
>
> In fact, MS-PCA can always distinguish  true covariance spikes \\(\\lambda_\\Sigma\\) and original contamination mean spikes \\(\\lambda_m\\) whenever they are simple or multiple or whenever they are close or seperated. Our goal is to detect \\(\\lambda_\\Sigma\\). Actually, MS-PCA will displace only \\(\\lambda_m\\)  after adding knockoff means, so we identify the covariance spikes as those ones that remain stable. Precisely, we have the following two situations:
>
> **Case 1** When  \\(\\lambda_\\Sigma\\) and \\(\\lambda_m\\) nearly coincide, MS‑PCA will displace  \\(\\lambda_m\\) away from \\(\\lambda_\\Sigma\\) after adding knockoff means, so that we can still distinguish them.
>
> **Case 2**   When \\(\\lambda_\\Sigma\\) and \\(\\lambda_m\\) are separated but MS-PCA will drive \\(\\lambda_m\\) close to \\(\\lambda_\\Sigma\\): in this case we can still distinguish them because only \\(\\lambda_m\\) is displaced while \\(\\lambda_\\Sigma\\) remains stable.
>
>
>
> **5. Finite Sample Behavior**
> It's well known that RMT concentration is very rapid (as illustrated in Figure 6), even for moderate \\(n\\). The theoretical guarantees typically remain reliable unless the sample size is very small. However, it's very challenging to derive exact non-asymptotic error bounds for eigenvalue perturbations. We will leave it to future research.
>
> **6. Correlated Contamination**
> Alignment of the mean‑shift direction with true signal eigenvectors is not problematic. After adding the additional mean‑shift perturbation, the existing mean‑shift direction changes and will no longer be aligned with the true signal eigenvectors, and the SVD and spectrum of the mean‑shift matrix change accordingly. Indeed these have been confirmed by a few experiments (not reported).
>
> **7. Comparison to Robust Covariance Estimators**
>
> We have conduct new experiments for comparison with several  methods:
> - RPCA-AAP: see related work in Section 1.1.
> - $\ell_1$-PCA:  see related work in Section 1.1.
> - Tyler's M estimator
> - Huber's M esimator
> - winsorized-PCA: winsorize the data before PCA
> - center-PCA: centering the data before PCA
>
> We provide results for the recovery of the largest principal component under mean‑shift Gaussian mixture contamination. Results show that MS‑PCA outperforms other methods in this setting. The source code for this experiment is available at the anonymous GitHub link provided in the paper. Please refer to Table 1 in the rebuttal to **Reviewer zV7x**, under the section “Additional Experiments for Broader Comparison.”
>
> ## Answers to Weaknesses
>
> We found that the Weaknesses Section is more or less a summary of the Questions Section and there is a huge overlap between the two. Therefore, we refer to the answers given above to the Questions Section.

---

> > ### Author Rebuttal · Reviewer_ByR5 · 2026-04-04
> >
> > We thank the reviewer for the thoughtful and detailed feedback. We appreciate the concerns regarding detectability under weak contamination, sensitivity to parameter selection, and applicability to real-world settings. We will clarify the below-threshold regime where MS-PCA becomes neutral, and better motivate practical choices of
> > 𝛾, including references to the spike-forward mapping. We also acknowledge the importance of real-data validation and will expand discussion on modeling limitations and ongoing extensions. Additionally, we will further clarify the behavior under closely spaced spikes, finite-sample regimes, and correlated contamination. Finally, we appreciate the suggestion to compare with robust covariance estimators and will incorporate the additional experimental results into the main paper for completeness.

---

### Official Review · Reviewer_56L3 · 2026-02-25

**Soundness:** 2
**Presentation:** 2
**Significance:** 3
**Originality:** 4
**Overall Recommendation:** 3
**Confidence:** 4

**Summary:**

This paper studies the PCA problem when the data is contaminated by noise that shift the mean of the original covariance estimates. The paper starts by empiirically show that robust PCA and regular PCA fails at such task, then provided theoretical insight that eigenvalues induced by the signal and the noise are decoupled. Based on this observation, the paper proposed the algorithm MS-PCA that solves the problem by further injecting noise into the matrix. Through experiment on synthetic data, the paper demonstrated that the proposed method's performance gain compared with robust PCA.

**Compliance With Llm Reviewing Policy:**

Affirmed.

**Final Justification:**

I would like to keep the score. Although some of my concerns are resolved in the rebuttal period, the question of experiment on real-world dataset is still not present, which I believe is an important component for me to recommend acceptance.

**Key Questions For Authors:**

Please explain how to choose $\mathbf{m]'$ in the algorithm.

**Limitations:**

yes

**Strengths And Weaknesses:**

**Strengths**

1. The paper tackles a significance problem of mean-shift PCA that previous method fails to solve.
2. The paper presented interesting theoretical guarantee that the eigenvalues spectrum of the contaminated matrix is well-separated into a part that correspond to the signal and a part that correspond to the added noise.
3. The paper provided a weaker theoretical guarantee going beyond Gaussian data.

**Weaknesses**

1. The presentation of the paper is unclear, especially in Section 1 and 2. In particular, the notation $\pi$ and $\theta$ is used extensively before formally introduced in Section 3. Also, algorithm 1 is not presented clearly. In particular, the choice of $\mathbf{m}'$ is crucial to the algorithm, but not explicitly defined. Lastly, in Notation 4 it should be noted that $\delta$ defines the Dirac measure or the Kronecker delta function..
2. Since $\mathbf{m}'$ is not clearly defined, Corollary 3.7 seems vague and cannot be fully justified by its current proof.
3. Conclusion of Theorem 3.11 seems to be weak. It only guarantees that the original eigenvector is still an eigenvector, but in the case of finding the top-$k$ eigenvectors, it is not clear whether the top-$k$ eigenvectors of the clean matrix is still the top-$k$ eigenvectors when contaminated.
4. Experimental results is quite weak since it lacks performance on real-world dataset.

---

> ### Author Rebuttal · Authors · 2026-03-30
>
> We thank Reviewer 56L3 for the constructive comments and for recognizing the originality of our contribution. We address the raised concerns below (**W** stands for Weakness, **Q** stands for Questions).
>
> ## Answer to Weakness
>
> **1. [W1] Notation problems**
>
> Thank you for raising this issue. We have revised all the notations such that they are well defined at their first appearance.
>
> **2. [W1,Q1] Definition and choice of  $\\mathbf{m}'$**
> We have now clarified that \\(A_n'=\\mathbf{m}'\gamma'^\top\\) in the definition of the **Perturbation Generation** paragragh on Page 3 where $\\mathbf{m}'$ is now well defined.
>
> Regarding the choice of  $\\mathbf{m}'$, our algorithm also depends on  the choice of  \\(\\pi'\\) and \\(\\theta'^2\\). In the revised paper,  we determine these three parameters as follows:
>
> - mixture weight \\(\\pi'\\) : \\(\\pi'\\) should be large enough. In practice we choose \\(\\pi'\\) from 0.5 to 1. In particular, all the experiments in our paper set \\(\\pi'\\) =1.
>
> - signal strength \\(\\theta'^2\\):  We recommend choosing \\(\\theta'\\) such that the newly generated spike(s) are of the same order as the existing spikes. In the Gaussian mixture case, one can use a closed‑form formula to compute the desired \\(\\theta'\\) via the inverse of the spike‑forward mapping \\(g\\) (the equation at line 161, right column, or the \\(D\\)-transform for the general case) to the currently observed spectral spike eigenvalue(s). Details can be found in Lines 156–164 (right column) in the paper.
>
>
> - direction  \\(\\mathbf{m}'\\): \\(\\pi'\\) and \\(\\theta'^2\\) determine the length \\(\\|\\mathbf{m}'\\|\\), while its direction \\(\\mathbf{v} = \\mathbf{m}'/\\|\\mathbf{m}'\\|\\) can be chosen randomly (Lines 148–150, right column) — either as a uniformly random vector on the unit sphere or as an i.i.d. Gaussian vector.
>
>
> **3. [W2] Corollary 3.7**
>
> We have clarified the definition of \\(\\mathbf{m}'\\), so Corollary 3.7 is also clearly justified.
>
> **4. [W3] Order of Top‑\\(K\\) Principal Components**
> Theorem 3.11 is indeed not enough on its own to guarrantee the recovery of the signal  eigenvector space. Our method is based on the combination of Theorem 3.11 and Theorem 3.5.  Specifically, Theorem 3.5 states that the ordering of the top‑\\(K\\) principal components changes after mean‑shift contamination, which is precisely why standard PCA fails under mean‑shift contamination. However, the eigenspace associated with the covariance signal remains stable under mean-shift perturbation, even if the ordering of the top‑\\(k\\) eigenvalues changes. MS‑PCA leverages this stability to recover the relevant components.
>
> **5. [W4] More Experiments**
>
> If the contamination in real data follows a mean‑shift pattern, our algorithm can address the problem with theoretical guarantees (under compact spectral support). However, real‑world noise may follow more complex patterns that can not be captured by mean-shift solely. That's why the paper doesn't contain real data analysis at the moment.  We are continuing to extend our modelling to catch up with real data examples.
>
> On the other hand,  we have improved our simulation study by additional comparisons with several robust PCA methods:
> - RPCA-AAP: see related work in Section 1.1.
> - $\ell_1$-PCA:  see related work in Section 1.1.
> - Tyler's M estimator
> - Huber's M esimator
> - winsorized-PCA: winsorize the data before PCA
> - center-PCA: centering the data before PCA
>
> We provide results for the recovery of the largest principal component under mean‑shift Gaussian mixture contamination. Our MS‑PCA outperforms other methods in this setting.  Please refer to Table 1 in the rebuttal to **Reviewer zV7x**, under the section “Additional Experiments for Broader Comparison.”

---

> > ### Author Rebuttal · Reviewer_56L3 · 2026-04-03
> >
> > My other concerns are resolved. The one question I have regarding the experiment is that, although real-world data is more complex, if mean-shift is a significant topic, wouldn't applying the algorithm at least result in some performance improvement?

---

> > > ### Author Response · Authors · 2026-04-04
> > >
> > > Thank you for your last comment.  Although the real world is more complex and the structure of the noise in contaminated data is unknown, we conjecture that as long as the noise contains a mean-shift component, our method, MS-PCA, can still partially eliminate the noise effect and achieve superior performance compared to vanilla PCA and most existing robust PCA methods. This conjecture is numerically validated in a newly added simulation experiment, in which the data is contaminated with noise containing both mean-shift and covariance-shift. Under this setting, our method exhibits the best performance. Further details are provided below.
> > >
> > > We provide recovery results  for the largest PC under mean‑shift Gaussian mixture contamination (larger is better).
> > >
> > > We add a covariance-wise perturbation: the data follow a mixture of $\mathcal{N}(0, \mathbf{I} + \mathbf{P}_1)$ and $\mathcal{N}(\mathbf{m}_1, (\mathbf{I} + \mathbf{P}_1)(\mathbf{I} + \mathbf{P}_2))$. Here $\pi_1$ is the mixture weight of the second component, and $\ell_2$ is the spike of the rank-1 matrix $\mathbf{P}_2$. Results based on 100 independent trials.
> > >
> > > As shown in the table, MS‑PCA largely outperforms all other methods in this setting when there is a mean shift in the data: our recovery rates are all above 95\% while others are all below 15\%.
> > >
> > > | (\\(\pi_1\\), \\(\ell_2\\)) | MS-PCA | RPCA-AAP | Tyler | Huber | \\(\ell_1\\)-PCA | winsorized-PCA | center-PCA |
> > > |------------------------|--------|----------|-------|-------|----------------|----------------|-----------|
> > > | \\((5\\%,\ 2)\\) | \\(97.01 \pm 8.01\\%\\) | \\(7.69 \pm 6.42\\%\\) | \\(8.67 \pm 7.26\\%\\) | \\(9.14 \pm 7.49\\%\\) | \\(13.13 \pm 10.61\\%\\) | \\(8.59 \pm 7.20\\%\\) | \\(8.64 \pm 7.32\\%\\) |
> > > | \\((5\\%,\ 4)\\) | \\(97.52 \pm 1.76\\%\\) | \\(9.09 \pm 7.90\\%\\) | \\(9.82 \pm 8.20\\%\\) | \\(9.83 \pm 8.37\\%\\) | \\(15.17 \pm 11.83\\%\\) | \\(9.85 \pm 8.19\\%\\) | \\(9.83 \pm 8.14\\%\\) |
> > > | \\((5\\%,\ 8)\\) | \\(97.15 \pm 8.37\\%\\) | \\(7.84 \pm 5.91\\%\\) | \\(8.58 \pm 6.64\\%\\) | \\(9.06 \pm 6.70\\%\\) | \\(13.46 \pm 10.19\\%\\) | \\(8.50 \pm 6.68\\%\\) | \\(8.54 \pm 6.72\\%\\) |
> > > | \\((5\\%,\ 16)\\) | \\(96.54 \pm 8.67\\%\\) | \\(9.74 \pm 10.52\\%\\) | \\(10.88 \pm 11.64\\%\\) | \\(11.35 \pm 11.92\\%\\) | \\(16.01 \pm 14.33\\%\\) | \\(10.79 \pm 11.56\\%\\) | \\(10.71 \pm 11.35\\%\\) |
> > > | \\((10\\%,\ 2)\\) | \\(97.52 \pm 1.45\\%\\) | \\(8.61 \pm 6.21\\%\\) | \\(10.29 \pm 7.37\\%\\) | \\(10.38 \pm 7.53\\%\\) | \\(13.32 \pm 10.05\\%\\) | \\(10.26 \pm 7.36\\%\\) | \\(10.26 \pm 7.36\\%\\) |
> > > | \\((10\\%,\ 4)\\) | \\(97.80 \pm 1.40\\%\\) | \\(7.42 \pm 6.46\\%\\) | \\(8.81 \pm 7.45\\%\\) | \\(9.16 \pm 7.54\\%\\) | \\(11.60 \pm 9.61\\%\\) | \\(8.74 \pm 7.45\\%\\) | \\(8.78 \pm 7.56\\%\\) |
> > > | \\((10\\%,\ 8)\\) | \\(97.64 \pm 1.60\\%\\) | \\(8.66 \pm 6.04\\%\\) | \\(10.09 \pm 7.56\\%\\) | \\(10.13 \pm 7.75\\%\\) | \\(13.55 \pm 10.04\\%\\) | \\(10.08 \pm 7.54\\%\\) | \\(10.04 \pm 7.45\\%\\) |
> > > | \\((10\\%,\ 16)\\) | \\(97.75 \pm 1.27\\%\\) | \\(8.06 \pm 5.77\\%\\) | \\(9.87 \pm 6.82\\%\\) | \\(10.02 \pm 6.97\\%\\) | \\(13.10 \pm 9.07\\%\\) | \\(9.83 \pm 6.82\\%\\) | \\(9.93 \pm 6.73\\%\\) |
> > > | \\((15\\%,\ 2)\\) | \\(96.63 \pm 9.72\\%\\) | \\(8.33 \pm 6.15\\%\\) | \\(10.74 \pm 7.71\\%\\) | \\(10.97 \pm 7.75\\%\\) | \\(12.20 \pm 9.01\\%\\) | \\(10.70 \pm 7.71\\%\\) | \\(10.79 \pm 7.62\\%\\) |
> > > | \\((15\\%,\ 4)\\) | \\(97.65 \pm 1.48\\%\\) | \\(7.90 \pm 6.09\\%\\) | \\(9.95 \pm 7.56\\%\\) | \\(9.98 \pm 7.60\\%\\) | \\(11.79 \pm 8.60\\%\\) | \\(9.95 \pm 7.58\\%\\) | \\(10.01 \pm 7.49\\%\\) |
> > > | \\((15\\%,\ 8)\\) | \\(97.45 \pm 1.29\\%\\) | \\(8.57 \pm 6.22\\%\\) | \\(10.81 \pm 7.77\\%\\) | \\(10.83 \pm 8.06\\%\\) | \\(12.51 \pm 9.10\\%\\) | \\(10.81 \pm 7.74\\%\\) | \\(10.98 \pm 7.77\\%\\) |
> > > | \\((15\\%,\ 16)\\) | \\(95.68 \pm 13.68\\%\\) | \\(8.97 \pm 6.83\\%\\) | \\(11.82 \pm 8.95\\%\\) | \\(11.93 \pm 8.98\\%\\) | \\(13.45 \pm 10.63\\%\\) | \\(11.79 \pm 8.95\\%\\) | \\(11.71 \pm 9.10\\%\\) |
> > > | \\((20\\%,\ 2)\\) | \\(97.91 \pm 1.11\\%\\) | \\(7.63 \pm 6.01\\%\\) | \\(11.18 \pm 8.22\\%\\) | \\(11.51 \pm 7.93\\%\\) | \\(11.17 \pm 8.49\\%\\) | \\(11.13 \pm 8.29\\%\\) | \\(11.21 \pm 8.16\\%\\) |
> > > | \\((20\\%,\ 4)\\) | \\(95.02 \pm 16.58\\%\\) | \\(8.64 \pm 6.41\\%\\) | \\(12.84 \pm 9.51\\%\\) | \\(12.74 \pm 10.07\\%\\) | \\(13.35 \pm 9.90\\%\\) | \\(12.89 \pm 9.40\\%\\) | \\(13.02 \pm 9.28\\%\\) |
> > > | \\((20\\%,\ 8)\\) | \\(96.76 \pm 9.82\\%\\) | \\(7.93 \pm 6.04\\%\\) | \\(11.33 \pm 9.16\\%\\) | \\(11.81 \pm 9.34\\%\\) | \\(11.63 \pm 9.30\\%\\) | \\(11.27 \pm 9.15\\%\\) | \\(11.41 \pm 9.22\\%\\) |
> > > | \\((20\\%,\ 16)\\) | \\(96.68 \pm 9.74\\%\\) | \\(7.84 \pm 5.91\\%\\) | \\(11.23 \pm 8.57\\%\\) | \\(11.24 \pm 8.60\\%\\) | \\(11.50 \pm 8.63\\%\\) | \\(11.24 \pm 8.58\\%\\) | \\(11.09 \pm 8.37\\%\\) |

---

### Official Review · Reviewer_3fe3 · 2026-03-06

**Soundness:** 2
**Presentation:** 2
**Significance:** 3
**Originality:** 3
**Overall Recommendation:** 4
**Confidence:** 4

**Summary:**

This paper studies principal component analysis (PCA) in the presence of outliers. In particular, the paper focuses on mean-shift contamination, which is a typical contamination in practice. In general, robust methods attempt to weaken the effect of contamination in order to obtain reliable inference. In contrast to these conventional approaches, the authors propose a novel alternative method: they intentionally add artificial mean-shift contamination and then detect the central PCA components that are independent of the contamination. The key idea is that artificial mean-shift contamination changes the eigenvalues associated with the contaminated components, while leaving the central eigenvalues unchanged.
The proposed method is supported by theoretical analysis, and numerical experiments are also provided.

**Compliance With Llm Reviewing Policy:**

Affirmed.

**Final Justification:**

The authors have additionally addressed some of my concerns in their second rebuttal. Therefore, I have decided to increase my score from 3 to 4. However, as mentioned in the review, there are many unclear descriptions. The authors should improve the clarity of the paper.

**Key Questions For Authors:**

Please refer to the section titled "Strengths and Weaknesses."

The following points are particularly important to address:

Literature Review

Clarity in the Theoretical Explanation

Theory

Comparison

**Limitations:**

The authors have not adequately discussed the limitations.
For details, see the section titled "Strengths and Weaknesses."

**Strengths And Weaknesses:**

The idea of adding artificial mean-shift contamination is very interesting. However, unfortunately, the paper contains many unclear descriptions, and the theoretical justification and experimental evaluation are not yet sufficient. In its current form, the paper does not meet the standard required for a top conference.

The comments are somewhat lengthy because the reviewer believes that this paper has strong potential. If the authors adequately address the concerns raised, I would be happy to increase the score.




Clarity

The paper is generally well written and easy to follow.
However, many notations are not explained in the main part of the paper (although some of them are explained in the Appendix), or they are explained only later in the text. As a result, when such notations first appear, it is difficult for readers to understand what the authors intend to convey. The following are just a few examples from page 3:
"The mixture weight $\pi'$ associated with $\gamma'$. $D_{\mu_\infty}^{-1}(\lambda^++\epsilon)$ (Remark C.2)". Some notations in this sentence are defined only in the Appendix. The authors refer to  “the underlying representation $\theta^2$”, but $\theta^2$ is not defined in the main text. Similarly, “the spike-forward mapping $g$, characterized by Proposition C.1,” is defined only in the Appendix. These are merely examples, but they illustrate a broader issue. The main part of the paper would ideally be self-contained so that readers can follow the arguments without repeatedly consulting the Appendix.
In addition, the authors write: “In practice, we recommend choosing ${\theta'}^2$ to be comparable in magnitude to the underlying representation $\theta^2$ of the observed spiked eigenvalues we aim to identify. For example, to target the largest observed eigenvalue $\tilde{\lambda}_1$, one may set ${\theta'}^2 = 2 g^{-1}(\tilde{\lambda}_1)$, where …”. Here the authors use the phrase “in practice,” but the function $g$ is not defined in the main text; it appears only in the Appendix. Therefore, readers cannot determine an appropriate value from the main part of the paper alone. Again, the paper would benefit from making the main text self-contained.


Reproducibility

Some necessary information is missing, making it difficult to reproduce certain results.
 In Figure 2, the simulation setting is not described in sufficient detail. For instance, what type of Gaussian distribution was used? What type of outlier distribution was considered? Without this information, the simulation results cannot be reproduced.
Similarly, in Figure 3, the sample size $n$ is not provided. Consequently, the illustrative example cannot be reproduced. In addition, the quantity $l_1$ is used here but is not defined until the next page.



Literature Review

In the Conclusion, the authors state: “Furthermore, the term ‘Robust PCA’ is arguably a misnomer—a point subtly acknowledged by the question mark in the title of the original Robust PCA paper by Candès et al. (2011). The prevailing formulation addresses the recovery of a low-rank matrix from sparse, large-magnitude noise, which diverges from the classical definition of robustness in statistics concerning resistance to contamination.”
The reviewer agrees with this observation. At the same time, another question arises: why do the authors not cite papers that focus on the classical statistical veiw of robustness? Such literature is not discussed at all.
For example, if one searches Google Scholar using the keywords “robust,” “PCA,” and “influence function” (where “influence function” is a typical concept in classical robustness), many relevant papers can be found.


Clarity in the Theoretical Explanation

In Notation 1, $\Lambda_A$ and $\Lambda_P$ are mentioned but are not defined clearly.  Consequently, the explanation of “Order of Fluctuation” is difficult to follow. Moreover, the description of this concept is not presented in a rigorous form. Since this paragraph appears to play a central role in the paper, the lack of a clear and formal explanation makes it difficult to understand the intended meaning of the order.
In addition, the authors set $\epsilon = O(n^{-1/2})$, but there is no clear explanation of why this choice works well, although some explanations are present.
The authors also state: “Empirically (see Figure 5), $C = 1$ suffices when $d$ is sufficiently large. For moderate $d$ and small aspect ratio $c < 1$, we choose $C = 1/c$ to compensate for the weaker concentration of measure in low dimensions.”
However, it is not clear how this explanation follows from Figure 5. In fact, since $c = 1$ in the figure, we have $1/c = 1$, so the reciprocal does not change the value. Moreover, the reviewer cannot understand the role of $O(d^{1/3})$ in the figure, because this order is not discussed anywhere in the paper.


Theory

Corollary 3.7 is correct. However, it does not seem sufficient to separate population covariance eigenvalues from mean-shift eigenvalues using Algorithm 1. The key issue appears in the proof of Corollary 3.7. Let $A_n= \sum_{i=1}^{k} m(i)\gamma^\top(i)$.
Assume that the vectors $m(i)$ are orthogonal. If $A_n' = m(1)\gamma^\top(1)$, many eigenvalues remain unchanged, and Algorithm 1 may accept many mean-shift eigenvalues. Therefore, Corollary 3.7 is not sufficient to explain why Algorithm 1 works well.
However, the probability of such a situation may be very small. Moreover, if the added mean-shift contamination is more complex, the eigenvalues may change significantly, and Algorithm 1 may perform better, as suggested by the experimental results. Based on this observation, the reviewer recommends that the authors develop a more informative theoretical result beyond Corollary 3.7.


Comparison

The proposed method is compared only with Accelerated Alternating Projections (AAP) introduced by Cai et al. (2019). The proposed method should ideally be compared with a wider range of methods, including classical methods. In particular, one drawback of the proposed method is that the value $C$ in the threshold $\varepsilon = C n^{-1/2}$ must be chosen manually. The authors state: “Empirically (see Figure 5), $C = 1$ suffices when $d$ is sufficiently large.” However, this conclusion is based on only one simulation model. In practical applications, a good method for selecting $C$ will be highly desirable.

---

> ### Author Rebuttal · Authors · 2026-03-30
>
> We thank Reviewer 3fe3 for the thorough and detailed comments. We are encouraged by the reviewer’s note on the paper’s strong potential, and we address each point carefully below.
>
> ## Response to Weakness
>
> **Clarity**
>
> Thank you very much for pointing out this issue. In the revised paper, we will revise all the notations and theoretical formulas such that all the notations are well defined at their first appearance.
>
> **Reproducibility**
> We have double checked the missing information raised by the reviewer. Precisely, the experiment of Figure 2 uses the same setup as Figure 6, whose experimental setting is explained in Section 4 (Experiment), paragraph “Comparison to Robust PCA approach.” The Gaussian used is the Two‑Component Gaussian Mixture with One‑Spike Population Covariance defined there. This is stated in Appendix E (Experiment and Implementation Detail), line 800: “All Gaussian experiments follow the same parameter configuration detailed in Section 4.”
>
> The parameter \\(n\\) for Figure 3 is \\(n=10^3\\). This will be specified in the revised version.
>
> We will also remove the sentence at lines 366–368 (left column): “We set the aspect ratio \\(c = 1\\), i.e., the dimension \\(d\\) is equal to the sample size \\(n\\),” as the aspect ratio \\(c\\) is varying in Figure 6.
>
> **Literature Review**
>
> Thank you for pointing out this issue. We have now added several relevant methods for classic robust PCA , including Tyler's M estimator, Huber's M estimator etc.
>
> - Tyler's M estimator [1,2]
> - Huber's M esimator [3]
> - winsorized-PCA: winsorize the data before PCA [3]
> - center-PCA: centering the data before PCA [3]
> - RPCA-AAP: see related work in Section 1.1.  [4]
> - $\ell_1$-PCA:  see related work in Section 1.1 [5]
>
> In addition, we have run new experiments comparing our method with these popular robust PCA methods.
>
> [1] Complex Elliptically Symmetric Distributions: Survey, New Results and Applications E. Ollila, D.E. Tyler, V. Koivunen, H.V. Poor. IEEE Transactions on Signal Processing, 2012.
>
> [2] A distribution-free M-estimator of multivariate scatter D.E. Tyler. The Annals of Statistics, 1987.
>
> [3] P. J. Huber, Robust Statistics. New York: Wiley, 1981.
>
> [4] http://jmlr.org/papers/v20/18-022.html
>
> [5] doi:10.1109/TSP.2014.2338077.
>
> **Clarity in the Theoretical Explanation**
>
> As explained before, all the notations will be revised and explained at their first appearance. In particular, the definition of  \\(\Lambda_A\\)  and  \\(\Lambda_P\\)  in Theorem 1 will be moved to earlier place.
>
> We will explain more clearly the choice of \\(\epsilon=O(n^{-1/2})\\).
>
> For the orders in Figure 5, it's clear that the fluctuations follow the trend of \\(O(d^{-1/2})\\).
> The order \\(O(d^{-2/3})\\) corresponds to the fluctuation order of eigenvalues at the edge:
> we have added the upper order \\(O(d^{-1/3})\\) for graphical comparison. We will revise the paper to make it more clear.
>
> **Theory**
> The spectral change is substantial even with a single artificial mean‑shift. For example, let \\(A_n = \\sum_{i=1}^k m_i\\gamma_i^\\top\\) and \\(A' = m'_1\\gamma_1'^\\top\\), where \\(m_i\\) are mutually orthogonal, and suppose we add a full additional mean‑shift with \\(\\pi_1'=1\\) i.e. \\(\gamma_1'=(1,\cdots,1)\\), with  \\(\\|m_1'\\|\\)  large enough (see Page 3 of the paper for detail),  then after adding \\(A'\\), the vectors \\(m_i + m_1'\\) are no longer mutually orthogonal, and their SVD and corresponding spectral spikes all change. Actually Corollary 3.7 covers all useful cases.
>
>
>
> **Comparison**
>
> 1. We have conduct new experiments for comparison with several  methods:
> - RPCA-AAP: see related work in Section 1.1.
> - $\ell_1$-PCA:  see related work in Section 1.1.
> - Tyler's M estimator
> - Huber's M esimator
> - winsorized-PCA: winsorize the data before PCA
> - center-PCA: centering the data before PCA
>
> We provide results for the recovery of the largest principal component under mean‑shift Gaussian mixture contamination. Results show that MS‑PCA outperforms other methods in this setting. The source code for this experiment is available at the anonymous GitHub link provided in the paper. Please refer to Table 1 in the rebuttal to **Reviewer zV7x**, under the section “Additional Experiments for Broader Comparison.”
>
>
>
> 2. Selection of Hyperparameter \\(C\\)
>
> This is a good question. Since our method is grounded in RMT, spectral concentration typically converges very rapidly as the dimension grows, and the threshold \\(C\\) is normally not a critical concern in high dimensions. In lower‑dimensional settings, a small manual search over \\(C\\) may be needed (by simulation in the same dimensional settings, or faking mean-shift noise then detect). However, given the low time complexity of MS‑PCA, this tuning incurs negligible cost.

---

> > ### Author Rebuttal · Reviewer_3fe3 · 2026-04-02
> >
> > The new experiments, including comparisons with traditional robust PCA methods, are convincing. It would further strengthen the paper if you could provide a more detailed explanation of why the traditional approaches do not perform well in this setting.
> >
> > The choice of $\epsilon=O(n^{-1/2})$ appears to be a key component of the method. Unfortunately, there is no additional explanation provided for this choice.
> >
> > Your explanation of Figure 5 is understandable, particularly regarding why the choice of
> > $C$ is not critical.
> >
> > You state that “the vectors $m_i$ and $m_1'$ are no longer mutually orthogonal.” However, as I pointed out previously, there are cases such as $m_1=m_1'$. In such cases, as explained in my earlier review, the method may not perform well.
> >
> > Unfortunately, I will keep my original score, as several critical questions remain insufficiently addressed.

---

> > > ### Author Response · Authors · 2026-04-02
> > >
> > > We thank the reviewer for approval of part of our previous revision, we now reply to the remaining questions.
> > >
> > > 1. The traditional approaches are designed for low-dimensional statistics settings, i.e., $d/n \to 0$, not for the high dimensional regimes when $d/n \sim c>0$. It's well known that traditional PCA is highly biased in high dimensions [1]. This is the fundamental reason why traditional approaches fail in identifying mean-shifts in high dimensional PCA. In contrary, RMT is capable of identifying the bias created by the high dimensions, then by a debiasing procedure we can obtain a consistent estimator of the mean-shift parameters.
> > > [1] Johnstone, I. M., and  Paul, D. (2018). PCA in high dimensions: An orientation. Proceedings of the IEEE, 106(8), 1277-1292.
> > >
> > > 2. The choice of $\epsilon=Cn^{-1/2}$ is determined by order of fluctuation of the population covariance spike eigenvalues $\tilde{\lambda}_i$'s, see the paragraph of **Order of Fluctuation** line 206-217 on Page 4. On the other hand, the changes in spike eigenvalues from the mean-shift part created by the knockoff means are of constant order $O(1)$. These two different orders allow us to identify the population covariance spike eigenvalues. Principal components corresponding to population covariance spike eigenvalues are our target. We will add these new explanation in the revised text.
> > >
> > > 3.  Actually, Our algorithm is based on the changes in eigenvalues created by knockoff means, and these changes are of different order for mean-shift spike eigenvalues and population covariance spike eigenvalues.
> > > The alignment  $m_1=m_1'$ is not a problem for our algorithm. In this case, this artificial mean-shift addition will directly change the spectrum of $A$. For example, supposing  $m_1 = m_1'$, $\gamma_1 = \gamma_1' $, so  $A = m_1 \gamma_1^\top = A'$, the original perturbation strength $\theta =\sqrt{\pi_1}\\|m_1\\| = \theta'$. Then after adding knockoff mean, $A + A' = 2 m_1 \gamma_1^\top$, and the perturbation strength changes to $\theta +  \theta'=2\sqrt{\pi_1}\\|m_1\\|$. Hence by Theorem 3.5, the corresponding mean spike eigenvalues will change significantly after adding knockoff mean,  which is easy to be detected. On the contrary, the spike eigenvalues of the population covariance  remain stable before and after the addition of the knockoff mean. This stability enables us to effectively distinguish the mean-shift spike eigenvalues from those of the population covariance, thereby allowing us to identify the corresponding principal components associated with the population covariance eigenvalues, which is our main target.

---

### Official Review · Reviewer_f6ch · 2026-03-13

**Soundness:** 2
**Presentation:** 2
**Significance:** 2
**Originality:** 4
**Overall Recommendation:** 4
**Confidence:** 4

**Summary:**

This paper studies the problem of recovering true principal components from high-dimensional data contaminated by mean-shift noise, where a fraction of samples come from distributions with shifted means but identical covariance. The authors observe that standard PCA is highly sensitive to such contamination, and that existing Robust PCA (RPCA) methods, designed for sparse corruption, are fundamentally mismatched for this structure. Using Random Matrix Theory (RMT), specifically the framework of Benaych-Georges & Nadakuditi (2012) for finite-rank perturbations of large random matrices, the authors prove two main results: (1) the spiked eigenvalues of the contaminated sample covariance matrix decompose into two asymptotically independent sets, one from the true covariance and one from the mean-shift contamination; and (2) eigenvectors of the uncontaminated covariance remain approximate eigenvectors of the contaminated covariance with $O_p(n−^{1/2})$ error. Based on this, they propose MS-PCA, a two-stage algorithm that runs PCA on the contaminated data, injects an artificial mean-shift perturbation, runs PCA again, and retains only eigenpairs pairs that remain stable across the two PCA runs. Experiments on synthetic Gaussian mixtures demonstrate improvement over RPCA.

**Compliance With Llm Reviewing Policy:**

Affirmed.

**Final Justification:**

I appreciate the response by the authors. The response adequately answers my questions. I retain my original score.

**Key Questions For Authors:**

See weaknesses section

**Limitations:**

yes

**Strengths And Weaknesses:**

Strengths:

1) The paper identifies a well-motivated problem with a genuine gap: mean-shift contamination is prevalent in practice (data aggregation from heterogeneous sources, batch effects), yet RPCA's sparsity assumptions are structurally mismatched.

2) I really liked the idea of further corruption to identify the corrupted mean-shift directions. I have previously seen such an idea in the context of learning a diffusion model with corrupted data - https://arxiv.org/pdf/2305.19256. Its is really nice to see this concept analyzed theoretically.

Weaknesses: I don't see any obvious weaknesses, so I will state some questions.

1) Several other relevant baselines are not compared against: robust mean estimation followed by standard PCA, $\ell_1$-PCA methods (which the paper discusses but does not compare against), trimmed/winsorized PCA, median-of-means PCA, or some baseline centering-then-PCA approaches that first estimate and remove the mixture means.

2) On the theoretical front, I am very curious to understand a detailed comparison to robust PCA algorithms under adversarial contamination, see e.g https://arxiv.org/pdf/2006.06980 and the papers' contribution relevant to this literature.

3) When a true covariance spike and a contamination parameter are such that the two families of spiked eigenvalues nearly coincide, the invariance check cannot distinguish them, right?

4) Can the authors elaborate on the dependence of their technique on the incoherence assumption on the coordinates of the mean-shift vectors? Practical corruptions may be very adversarial, right?

5) Given that you cite Elhaik (2022) on PCA bias in population genetics as motivation, have you attempted to apply MS-PCA to actual population genetics datasets (e.g., the Human Genome Diversity Project)?

---

> ### Author Rebuttal · Authors · 2026-03-30
>
> We thank Reviewer f6ch for the positive assessment and the insightful questions. We address each point below.
>
> ## Answer to Questions (in the Weakness Section)
>
> **1. Comparison**
>
> Thank you for your suggestion. We have now conducted new experiments for comparison with several  methods:
> - RPCA-AAP: see related work in Section 1.1.
> - $\ell_1$-PCA:  see related work in Section 1.1.
> - Tyler's M estimator
> - Huber's M esimator
> - winsorized-PCA: winsorize the data before PCA
> - center-PCA: centering the data before PCA
>
> We provide results for the recovery of the largest principal component under mean‑shift Gaussian mixture contamination. As shown in the table, MS‑PCA outperforms other methods in this setting. The source code for this experiment is available at the anonymous GitHub link provided in the paper.
>
> **Table 1**: Alignment of the Largest Principal Component under different mixture proportions. Statistics (mean ± std) as percentages, calculated from 200 independent trials.
> \\(d=900,\\ n=10^3\\). The (hyper-)parameter settings are kept the same as in the experiments of Figure 6. Alignment is calculated as cosine similarity between 2 vectors, ranged in  $[0,1]$, **higher is better**.
>
> | \\(\\pi_1\\) | MS-PCA | RPCA-AAP | Tyler | Huber | \\(\\ell_1\\)-PCA | winsorized-PCA | center-PCA |
> |----------|--------|----------|-------|-------|------------|----------------|-----------|
> | 5% | \\(95.85 \\pm 12.53\\) | \\(8.40 \\pm 6.87\\) | \\(9.33 \\pm 7.46\\) | \\(9.72 \\pm 7.84\\) | \\(14.01 \\pm 10.73\\) | \\(9.26 \\pm 7.42\\) | \\(9.27 \\pm 7.37\\) |
> | 10% | \\(97.16 \\pm 6.96\\) | \\(8.75 \\pm 6.71\\) | \\(10.67 \\pm 8.00\\) | \\(10.95 \\pm 8.11\\) | \\(14.25 \\pm 10.46\\) | \\(10.63 \\pm 8.01\\) | \\(10.64 \\pm 8.10\\) |
> | 15% | \\(97.39 \\pm 7.03\\) | \\(7.47 \\pm 6.13\\) | \\(10.24 \\pm 8.12\\) | \\(10.51 \\pm 8.22\\) | \\(12.06 \\pm 9.33\\) | \\(10.22 \\pm 8.09\\) | \\(10.36 \\pm 8.07\\) |
> | 20% | \\(96.17 \\pm 11.82\\) | \\(7.74 \\pm 5.67\\) | \\(11.46 \\pm 8.63\\) | \\(11.53 \\pm 8.72\\) | \\(11.85 \\pm 8.81\\) | \\(11.47 \\pm 8.63\\) | \\(11.70 \\pm 8.70\\) |
>
> **2. Adversarial contamination**
>
> Thank you for suggesting comprison to robust PCA algorithms under adversarial contamination. We found  that the framework is quite different from mean-shift contamination and direct comparison is not straightforward. However, we will look into some simple comparison in the revised paper with adequate references (including https://arxiv.org/pdf/2006.06980).
>
>
>
> **3. When Two Spiked Eigenvalues Coincide**
>
> In fact, MS-PCA can always distinguish a  true covariance spike \\(\\lambda_\\Sigma\\) and an original contamination mean spike \\(\\lambda_m\\).
>
> **Case 1** When  \\(\\lambda_\\Sigma\\) and \\(\\lambda_m\\) nearly coincide, MS‑PCA will displace  \\(\\lambda_m\\) away from \\(\\lambda_\\Sigma\\) after adding knockoff means, so that we can still distinguish them.
>
> **Case 2**   When \\(\\lambda_\\Sigma\\) and \\(\\lambda_m\\) are separated but MS-PCA will drive \\(\\lambda_m\\) close to \\(\\lambda_\\Sigma\\): in this case we can still distinguish them because only \\(\\lambda_m\\) is displaced while \\(\\lambda_\\Sigma\\) remains stable.
>
>
> **4. Incoherence Assumption on Mean‑Shift**
> Our current theoretical results rely on the assumption that the mean vectors follow a certain uniform distribution. However, in practice, we have conducted experiments showing that this assumption is not strictly necessary. For example, even when we used fixed mean vectors (similar to adversarial attacks), the performance of our MS-PCA method remained unaffected. Proving this robustness theoretically remains an open question.
>
> **5. Genetics Data**
> Population genetics datasets typically have \\(d\\gg n\\), placing them in the ultra‑high‑dimensional regime. Our RMT tools are based on the high‑dimensional assumption \\(d=O(n)\\), which does not cover the ultra‑high‑dimensional case \\(d > O(n^\\alpha)\\). (The low‑dimensional case \\(d=O(1)\\) can be handled by existing robust PCA methods.) Hence the assumptions underlying our theoretical guarantees are not satisfied for such datasets.

---

> > ### Author Rebuttal · Reviewer_f6ch · 2026-04-03
> >
> > I appreciate the response by the authors. The response adequately answers my questions. I retain my original score.

---

> > > ### Author Response · Authors · 2026-04-04
> > >
> > > We are grateful that the reviewer found our response helpful, and we thank the reviewer for their professional and constructive feedback.

---

### Official Review · Reviewer_zV7x · 2026-03-13

**Soundness:** 2
**Presentation:** 3
**Significance:** 1
**Originality:** 2
**Overall Recommendation:** 3
**Confidence:** 4

**Summary:**

The paper proposes Mean-Shift PCA (MS-PCA), a two-stage PCA procedure designed to recover principal components under mean-shift contamination in high-dimensional mixture models. The method leverages random matrix theory (RMT) to argue that contamination-induced spikes and intrinsic covariance spikes are asymptotically independent. Based on this observation, the authors introduce an artificial knockoff mean-shift perturbation and identify stable eigenvalues via an invariance test. Theoretical guarantees are provided under spiked covariance models and more general compactly supported spectral distributions.

**Compliance With Llm Reviewing Policy:**

Affirmed.

**Final Justification:**

I thank the authors for providing the runtime. Overall a borderline paper.

**Key Questions For Authors:**

What is the end-to-end computational complexity and memory footprint of MS-PCA when implemented with iterative eigensolvers? How does it scale with $d, n, k$ and what is the practical runtime on realistically large datasets?

How sensitive is performance to the choice of knockoff mixture weight $\pi'$, direction $m'$, and strength $\theta'^2$?

**Limitations:**

While the paper has a clean theoretical narrative and addresses a real failure mode of PCA, I do not find the method compelling in its current form for ICML due to practical scalability limitations (two expensive PCA passes) and insufficient empirical comparison to modern robust PCA/subspace learning methods.

**Strengths And Weaknesses:**

Strengths:

This manuscript discusses an important aspect of robust high-dimensional learning—namely, how first-moment (mean) contamination can mislead PCA and how to exploit spectral phenomena to recover the uncontaminated subspace. It uses random matrix theory to argue a spectral decoupling between covariance-induced spikes and mean-shift-induced spikes, and proposes a practical algorithm (MS-PCA) that injects an additional knockoff mean-shift and performs a second PCA to identify which leading eigenvalues are stable under this perturbation. The paper leverages well-established additive low-rank perturbation theory to motivate eigenvalue separation and a stability test.

Weaknesses:

While theoretically appealing, the knockoff mean procedure is computationally expensive. Computing eigenvalues, even with iterative methods, can be very costly. The proposed method doubles this cost by design, and additionally requires a matching step across eigenvalues. This is not a minor implementation detail, the paper’s practical claim is that it uses only standard PCA operations, but standard PCA itself is the bottleneck at the relevant scales. As a result, the approach appears unlikely to work on genuinely large-scale/high-dimensional datasets where robustness is most needed.

The experimental comparison is narrow: it mainly benchmarks against a specific RPCA implementation (AAP) in a synthetic Gaussian setting. This does not establish competitiveness against modern robust PCA or robust subspace estimation methods that are designed with scalability and robustness tradeoffs in mind.

Algorithm 1 recommends setting the knockoff strength $\theta'^2$ to be comparable to targeted observed spikes (e.g., via $2g^{-1}(\tilde{\lambda_1}). This introduces additional dependencies on estimating spike mappings or thresholds and selecting perturbation magnitudes. In practice, misspecifying these choices could lead to either insufficient movement of mean-shift spikes (false stability) or unintended perturbation of the spectrum (over-rejection). The paper does not provide robust guidance on sensitivity to these choices, nor strong empirical evidence across diverse settings.

---

> ### Author Rebuttal · Authors · 2026-03-30
>
> We thank Reviewer zV7x for the careful reading and constructive feedback. We address the main concerns below.
>
> ## **Key Questions**
>
> **1. Computational Complexity**
>
> As suggested by Algorithm 1 (lines 167–168, right panel), we only compute the largest \\(K\\) principal components (which are spikes in the spectrum). Modern PCA implementations compute the top \\(K\\) principal components directly via truncated SVD or iterative eigensolvers (e.g., power iteration (Lanczos algorithm), randomized SVD), with the following complexity:
>
> - **Randomized SVD**:
>   \\[ O(n d \\log K) + O((n+d)K^2) \\]
>   In practice, for large datasets this simplifies to \\(O(K n d)\\).
>
> - **Power iteration (Lanczos)**:   \\[ O(K n d). \\]
>
>  In practice, the number of informative spikes is finite, i.e., \\(K = O(1)\\). **Hence the computational complexity of MS‑PCA is \\(O(nd)\\)**, which is the smallest complexity we can expect.
>
> **2. Sensitivity**
> The performance of the algorithm MS-PCA is indeed sensitive to the choice of  \\(\\pi'\\) and \\(\\theta'^2\\).
> However, based on our extensive experiments, we have found a working set of parameters as follows:
>
> - mixture weight \\(\\pi'\\) : \\(\\pi'\\) should be large enough. In practice we choose \\(\\pi'\\) from 0.5 to 1. In particular, all the experiments in our paper set \\(\\pi' =1\\).
>
> - signal strength \\(\\theta'^2\\):   If \\(\\theta'\\) is too small, no new spectral spike is generated.  We recommend choosing \\(\\theta'\\) such that the newly generated spike(s) are of the same order as the existing spikes. In the Gaussian mixture case, one can use a closed‑form formula to compute the desired \\(\\theta'\\) via the inverse of the spike‑forward mapping \\(g\\) (the equation at line 161, right column, or the \\(D\\)-transform for the general case) to the currently observed spectral spike eigenvalue(s). Details can be found in Lines 156–164 (right column) in the paper.
>
> - direction  \\(\\mathbf{m}'\\): \\(\\pi'\\) and \\(\\theta'^2\\) determine the length \\(\\|\\mathbf{m}'\\|\\), while its direction \\(\\mathbf{v} = \\mathbf{m}'/\\|\\mathbf{m}'\\|\\) can be chosen randomly (Lines 148–150, right column). Indeed the algorithm is insensitive to the choice of \\( \\mathbf{v}\\).
>
>
> ## **Weakness**
>
> **1.  Computation complexity**
>
> The computation complexity of our method is \\(O(nd)\\), which is the smallest complexity we can expect, see the answer to Question 1 above.
>
>
> **2. Additional Experiments for Broader Comparison**
> We have conducted new experiments for comparison with several  methods:
> - RPCA-AAP: see related work in Section 1.1.
> - $\ell_1$-PCA:  see related work in Section 1.1.
> - Tyler's M estimator
> - Huber's M esimator
> - winsorized-PCA: winsorize the data before PCA
> - center-PCA: centering the data before PCA
>
> We provide results for the recovery of the largest principal component under mean‑shift Gaussian mixture contamination. As shown in the table, MS‑PCA outperforms other methods in this setting. The source code for this experiment is available at the anonymous GitHub link provided in the paper.
>
> **Table 1**: Alignment of the Largest Principal Component under different mixture proportions. Statistics (mean ± std) as percentages, calculated from 200 independent trials.
> \\(d=900,\\ n=10^3\\). The (hyper-)parameter settings are kept the same as in the experiments of Figure 6. Alignment is calculated as cosine similarity between 2 vectors, ranged in  $[0,1]$, **higher is better**.
>
> | \\(\\pi_1\\) | MS-PCA | RPCA-AAP | Tyler | Huber | \\(\\ell_1\\)-PCA | winsorized-PCA | center-PCA |
> |----------|--------|----------|-------|-------|------------|----------------|-----------|
> | 5% | \\(95.85 \\pm 12.53\\) | \\(8.40 \\pm 6.87\\) | \\(9.33 \\pm 7.46\\) | \\(9.72 \\pm 7.84\\) | \\(14.01 \\pm 10.73\\) | \\(9.26 \\pm 7.42\\) | \\(9.27 \\pm 7.37\\) |
> | 10% | \\(97.16 \\pm 6.96\\) | \\(8.75 \\pm 6.71\\) | \\(10.67 \\pm 8.00\\) | \\(10.95 \\pm 8.11\\) | \\(14.25 \\pm 10.46\\) | \\(10.63 \\pm 8.01\\) | \\(10.64 \\pm 8.10\\) |
> | 15% | \\(97.39 \\pm 7.03\\) | \\(7.47 \\pm 6.13\\) | \\(10.24 \\pm 8.12\\) | \\(10.51 \\pm 8.22\\) | \\(12.06 \\pm 9.33\\) | \\(10.22 \\pm 8.09\\) | \\(10.36 \\pm 8.07\\) |
> | 20% | \\(96.17 \\pm 11.82\\) | \\(7.74 \\pm 5.67\\) | \\(11.46 \\pm 8.63\\) | \\(11.53 \\pm 8.72\\) | \\(11.85 \\pm 8.81\\) | \\(11.47 \\pm 8.63\\) | \\(11.70 \\pm 8.70\\) |
>
>
> **3. Sensitivity of parameters**
> Please refer to our answer to Question 2.
>
>
> ## **Limitations**
> We regret that our original article did not provide a sufficiently comprehensive explanation. We would like to clarify that the computational complexity of our algorithm is  \\( O(K n d) \\), which means it is highly scalable.
>
> Additionally, we have included new experiments comparing our method with more robust PCA approaches, and our method achieves the best performance. Please refer to our previous response for the detailed results.
>
> We hope these two improvement makes our paper more acceptable for ICML.

---

> > ### Author Rebuttal · Reviewer_zV7x · 2026-04-01
> >
> > The authors discuss computational complexity; however, they do not provide any empirical runtime results for their method compared to the new state-of-the-art algorithms. Given that the approach involves two computationally intensive PCA passes, the practical cost could be substantial, potentially undermining the overall motivation of the work.

---

> > > ### Author Response · Authors · 2026-04-01
> > >
> > > | Method | Runtime (1 PC, d=900,n=1000) | Runtime (9 PCs,d=900,n=1000) |
> > > |--------|----------------|-----------------|
> > > | MS-PCA | \\(14.2 \\pm 0.0137\\) ms | \\(19.2 \\pm 0.131\\) ms |
> > > | RPCA-AAP | \\(79.0 \\pm 0.176\\) ms | \\(99.6 \\pm 0.210\\) ms |
> > > | Tyler | \\(172 \\pm 0.159\\) ms | \\(174 \\pm 0.351\\) ms |
> > > | Huber | \\(3860 \\pm 4.37\\) ms | \\(3860 \\pm 8.5\\) ms |
> > > | \\(\\ell_1\\)-PCA | \\(8230 \\pm 294\\) ms | N/A (did not converge) |
> > >
> > >
> > >
> > > Following the reviewer's recommendation, we provide empirical runtimes for the experiments in Table 1 above. We compare our method to RPCA-AAP, a state-of-art method, and three robust PCA methods (Tyler, Huber, \\(\\ell_1\\)-PCA) suggested by other reviewers. The first column shows the time to compute the largest principal component, and the second column the time for the largest nine principal components. Measurements were performed on an AMD EPYC 9755 256‑core server (dual‑socket, 128 cores per socket) without GPU acceleration. Times are given as mean ± standard deviation, calculated by `%timeit`. The source code is available via the anonymous GitHub link.
> > >
> > > Our algorithm is more than five times faster than RPCA-AAP and substantially faster than the three other robust PCA methods.
> > >
> > > The following is the table comparing runtime for computing the largest principal component (1 PC) of MS‑PCA and SOTA RPCA (AAP) across increasing dimensions. All times are reported in milliseconds (ms) for consistency, with the mean ± standard deviation. All experiments were conducted with the aspect ratio fixed at \\(c = d/n = 1\\), i.e., the sample size \\(n\\) equals the dimension \\(d\\) following the settings described in the Section 4 Experiment.
> > > | Method | \\(d = 1000\\) | \\(d = 2000\\) | \\(d = 3000\\) | \\(d = 4000\\) | \\(d = 5000\\) | \\(d = 10000\\) |
> > > |--------|----------------|----------------|----------------|----------------|----------------|-----------------|
> > > | MS-PCA | \\(15.9 \\pm 0.0109\\) ms | \\(49.2 \\pm 0.0795\\) ms | \\(102 \\pm 0.22\\) ms | \\(216 \\pm 1.1\\) ms | \\(237 \\pm 0.516\\) ms | \\(832 \\pm 1.26\\) ms |
> > > | RPCA-AAP | \\(86 \\pm 0.102\\) ms | \\(369 \\pm 0.945\\) ms | \\(1050 \\pm 3.01\\) ms | \\(2490 \\pm 4.39\\) ms | \\(2750 \\pm 6.78\\) ms | \\(10500 \\pm 21.6\\) ms |
> > >
> > > We see than when the dimension \\(d\\) increases from 1000 to 10000, the run time ratio of the RPCA-AAP method with respect to our MS-PCA method increases from 5.4 to 12.6 approximately.

---

### Decision · Program_Chairs · 2026-04-30

**Decision:**

Accept (regular)

**Comment:**

The paper studies the elimination of mean-shift noisy components from PCA, establishing an original framework and strong theoretical foundation. The proposed MS-PCA method is elegant, which uses a "knockoff" perturbation to distinguish between population covariance spikes and mean-shift-induced spikes. The theoretical analysis based on random matrix theory is solid. The reviewers initially pointed out weaknesses in empirical validations and clarity, most of which have been addressed by the author(s) in their rebuttals. They should be incorporated in the revision.  Overall, the paper makes valuable contributions and should be accepted.